# HP1α targets the chromosomal passenger complex for activation at heterochromatin before mitotic entry

Jan G Ruppert[1], Kumiko Samejima[1], Melpomeni Platani[1] (iD), Oscar Molina[1,†], Hiroshi Kimura[2],
A Arockia Jeyaprakash[1] (iD), Shinya Ohta[3] & William C Earnshaw[1,*] (iD)

## Abstract

The chromosomal passenger complex (CPC) is directed to centromeres during mitosis via binding to H3T3ph and Sgo1. Whether and how heterochromatin protein 1α (HP1α) influences CPC localisation and function during mitotic entry is less clear. Here, we alter HP1α dynamics by fusing it to a CENP-B DNA-binding domain. Tethered HP1 strongly recruits the CPC, destabilising kinetochore–microtubule interactions and activating the spindle assembly checkpoint. During mitotic exit, the tethered HP1 traps active CPC at centromeres. These HP1-CPC clusters remain catalytically active throughout the subsequent cell cycle. We also detect interactions between endogenous HP1 and the CPC during $G_2$. HP1α and HP1γ cooperate to recruit the CPC to active foci in a CDK1-independent process. Live cell tracking with Fab fragments reveals that H3S10ph appears well before H3T3 is phosphorylated by Haspin kinase. Our results suggest that HP1 may concentrate and activate the CPC at centromeric heterochromatin in $G_2$ before Aurora B-mediated phosphorylation of H3S10 releases HP1 from chromatin and allows pathways dependent on H3T3ph and Sgo1 to redirect the CPC to mitotic centromeres.

**Keywords** CENP-B; chromosomal passenger complex; heterochromatin protein 1; histone H3S10 phosphorylation
**Subject Categories** Cell Cycle; Chromatin, Epigenetics, Genomics & Functional Genomics
**The EMBO Journal (2018) 37: e97677**

## Introduction

The chromosomal passenger complex (CPC) is a key regulatory factor that controls chromosome segregation during mitosis. The CPC regulates chromosome condensation, release of erroneous kinetochore–microtubule attachments, spindle assembly checkpoint (SAC) activation and cytokinesis [reviewed by (Carmena et al, 2012; van der Waal et al, 2012)]. A defining feature of the CPC is that it functions at different locations at different times during mitosis (Earnshaw & Bernat, 1991).

The CPC localises to centromeres in order to regulate chromosomal attachments to the mitotic spindle. A breakthrough came following the discovery of H3T3 phosphorylation by Haspin kinase (Dai et al, 2005) when it was realised that CPC localisation to mitotic centromeres involves survivin binding to H3T3ph (Kelly et al, 2010; Wang et al, 2010). Sgo1 targeting to H2AT120ph (a product of Bub1 kinase activity) provides a second mechanism for CPC targeting, with Haspin/H3T3ph localising the CPC to inner centromeres and Bub1/H2AT120ph localising it to kinetochores (Yamagishi et al, 2010).

Heterochromatin protein 1α (HP1α) was the first known binding partner of the CPC (Ainsztein et al, 1998). A C-terminal chromoshadow domain (CSD) mediates HP1 dimerisation and interaction with binding partners containing a PxVxL/I motif (Brasher et al, 2000; Smothers & Henikoff, 2000; Nozawa et al, 2010). These partners include the CPC proteins INCENP and Borealin (Ainsztein et al, 1998; Nozawa et al, 2010; Kang et al, 2011; Liu et al, 2014).

Here, we have examined the role of HP1 in CPC localisation and activation in HeLa cells. The three isoforms HP1α, HP1β and HP1γ bind to histone H3 di- or trimethylated on lysine 9 (H3K9me2/3) via conserved N-terminal chromo domains (CD; Bannister et al, 2001; Lachner et al, 2001). Aurora B kinase, the catalytic component of the CPC, regulates this interaction during mitosis by phosphorylating the adjacent Serine10 residue (H3S10ph), thereby releasing HP1 from chromatin (Fischle et al, 2005; Hirota et al, 2005). The reason for this release is unknown.

A recent report described cell lines with decreased levels of HP1 at mitotic centromeres (Abe et al, 2016). Decreased levels of HP1-bound CPC led to reduced Aurora B activity in vitro and in vivo and increased chromosome segregation errors. Surprisingly, HP1 overexpression was not sufficient to rescue accurate chromosome segregation in those cell lines (Abe et al, 2016).

1 Wellcome Trust Centre for Cell Biology, University of Edinburgh, Edinburgh, UK
2 Cell Biology Unit, Institute of Innovative Research, Tokyo Institute of Technology, Yokohama, Japan
3 Department of Biochemistry, Medical School, Kochi University, Nankoku, Kochi, Japan
*Corresponding author. Tel: +44 131 650 7101; E-mail: bill.earnshaw@ed.ac.uk
†Present address: Josep Carreras Leukaemia Research Institute, School of Medicine, University of Barcelona, Barcelona, Spain

To explore interactions between HP1 and the CPC at centromeres in a simplified system without the normal epigenetic regulation, we recruited HP1α constitutively by tethering it via the DNA-binding domain (DBD) of CENP-B. This binds a 17 bp "CENP-B box" motif within the alpha-satellite repeats of human centromeres (Earnshaw *et al*, 1987; Masumoto *et al*, 1989).

Tethered HP1α recruits the CPC to centromeres strongly enough to cause a mitotic arrest and over-ride normal CPC transfer to the central spindle at anaphase. The trapped centromeric CPC clusters remain active throughout the subsequent interphase. In order to investigate normal interactions between endogenous HP1 and the CPC during interphase, we characterised CPC activation and localisation during $G_2$ arrest in CDK1-as cells and HP1 knockout cell lines. H3S10ph (a reporter of CPC activity) appears in nuclear foci that co-localise with HP1α. These foci are missing in cells lacking HP1α + HP1γ. $G_2$ arrested cells with the H3S10ph foci are negative for H3T3ph. Indeed, live cell analysis shows that H3T3ph only appears much later, as cells enter prophase. Our results suggest that CPC recruitment to centromeres by H3T3ph occurs downstream of an initial localisation and activation of the CPC to heterochromatin mediated by HP1.

## Results

### Tethering HP1α to centromeres via fusion to a CENP-B DNA-binding domain causes a mitotic delay

In order to investigate the effect of altering HP1α dynamics at centromeres, we generated a fusion construct linking HP1α to the DNA-binding domain of CENP-B (CB; Pluta *et al*, 1992) and enhanced yellow fluorescent protein (EYFP/EY), referred as CB-EY-HP1α (Fig 1A), and transiently expressed this in HeLa cells. Tethering HP1α to centromeres led to an accumulation of cells in metaphase with a concomitant depletion of cells in anaphase and telophase (Fig 1B and C). Importantly, cells expressing EYFP-labelled HP1α (EY-HP1α, i.e. lacking the CENP-B targeting domain) did not show an altered mitotic progression compared to controls expressing only EYFP.

To modulate the strength of HP1α tethering, we mutated the CENP-B DNA-binding domain by substituting residues S40, N120 and R125 with alanine (CB^mut-EY-HP1α). These residues make specific contacts with the DNA and are conserved among different species (Fig EV1C and D; Tanaka *et al*, 2001). This mutant construct did not cause cells to accumulate in mitosis (Fig 1B and C).

All constructs were expressed at levels comparable to or less than that of endogenous HP1 (Fig 1D). We determined the dynamics of the various EYFP-tagged HP1α constructs using fluorescence recovery after photobleaching (FRAP; Fig 1E). Consistent with previous observations, the mean half-time of recovery ($t_{1/2}$) for EY-HP1α was 3.1 s (Schmiedeberg *et al*, 2004). In contrast, the recovery half-time of CB-EY-HP1α was ~49 s, a > 15-fold increase. Mutation of the CENP-B DNA-binding domain resulted in almost threefold faster dynamics with a $t_{1/2}$ for CB^mut-EY-HP1α of 18 s. These results suggested that slowing HP1α dynamics at centromeres might be responsible for the metaphase delay phenotype.

Heterochromatin protein 1 tethered using either the wild-type or mutated CENP-B DBD co-localised with untethered EY-HP1α in the inner centromere of prometaphase cells where centromeres are not under tension (Fig EV1A). In metaphase cells, where centromeres are stretched, the tethered HP1α split into two peaks that tracked the separating kinetochores, while untethered EY-HP1α remained as a single, somewhat broader, peak in the inner centromere (Fig EV1B). The tethered CB-EY-HP1α remained 0.2–0.3 μm internal to the peak of CENP-C, suggesting that it occupies a kinetochore-proximal domain, as previously described for CB-INCENP (Liu *et al*, 2009; Wang *et al*, 2011; Hengeveld *et al*, 2017).

We hypothesised that the mitotic delay induced by tethering HP1α to centromeres might be a consequence of constitutive retention of a mitotic regulator capable of interacting with HP1α. To test this hypothesis, we generated CB-EY-HP1α mutants carrying point mutations previously described to perturb different functions of HP1α (Fig 1F). The V22M mutation in the chromodomain (CD) prevents HP1α binding to H3K9me2/3 (Bannister *et al*, 2001; Lachner *et al*, 2001; Nielsen *et al*, 2001). The chromoshadow domain mutation I165E disrupts HP1 dimerisation, while W174A disrupts formation of the hydrophobic pocket required for PxVxL motif binding. The I165E and W174A mutations both block HP1α association with PxVxL motif-containing HP1 client proteins (Brasher *et al*, 2000; Thiru *et al*, 2004; Nozawa *et al*, 2010).

Immunoblotting analysis confirmed that all mutant proteins were expressed to a similar level (Fig EV1F). The localisation of tethered CB-EY-HP1α^W174A resembled that of CB-EY-HP1α, but CB-EY-HP1α^I165E was slightly more diffuse (Fig EV1E). FRAP analysis revealed that introducing the I165E mutation into CB-EY-HP1α, which prevents dimer formation in HP1, results in a $t_{1/2}$ of recovery of 8 s (Fig 1E). This is similar to the $t_{1/2}$ of CB-EY (6.8 s), which consists of only the DNA-binding domain and EYFP without any

**Figure 1. Tethering HP1α to centromeres via fusion to a CENP-B DNA-binding domain causes a mitotic delay.**

A  Schematic representation of the various HP1α constructs.
B  Frequency of mitotic HeLa cells 24 h after transfection with the indicated constructs (transfection efficiency ~70%, judged by fluorescence microscopy). Graphs represent the mean, and error bars represent the standard deviation (SD) of three independent experiments ($n$ = 500 cells per experiment).
C  Frequency of mitotic phases in HeLa cells 24 h after transfection. Graphs show mean and SD of three independent experiments ($n$ = 60 mitotic cells per experiment).
D  Immunoblot of HeLa cell lysates transfected with the indicated HP1α fusion constructs. HP1α fusion constructs and endogenous HP1α were identified using an anti-HP1α antibody, and GAPDH was used as a loading control.
E  Quantitative FRAP analysis of the indicated HP1α fusion constructs in interphase HeLa cells. Error bars show SD.
F  Diagram of the tethered HP1α mutants and their perturbed functions.
G  Frequency of mitotic phases in HeLa cells 24 h after transfection. Graphs show mean and SD of three independent experiments ($n$ = 60 mitotic cells per experiment).

Data information: (B, C, G) Statistical significance was determined by Fisher's exact test followed by the Benjamini–Hochberg multiple comparison test. *$P$ < 0.05; ***$P$ < 0.001; ****$P$ < 0.0001; n.s., not significant.

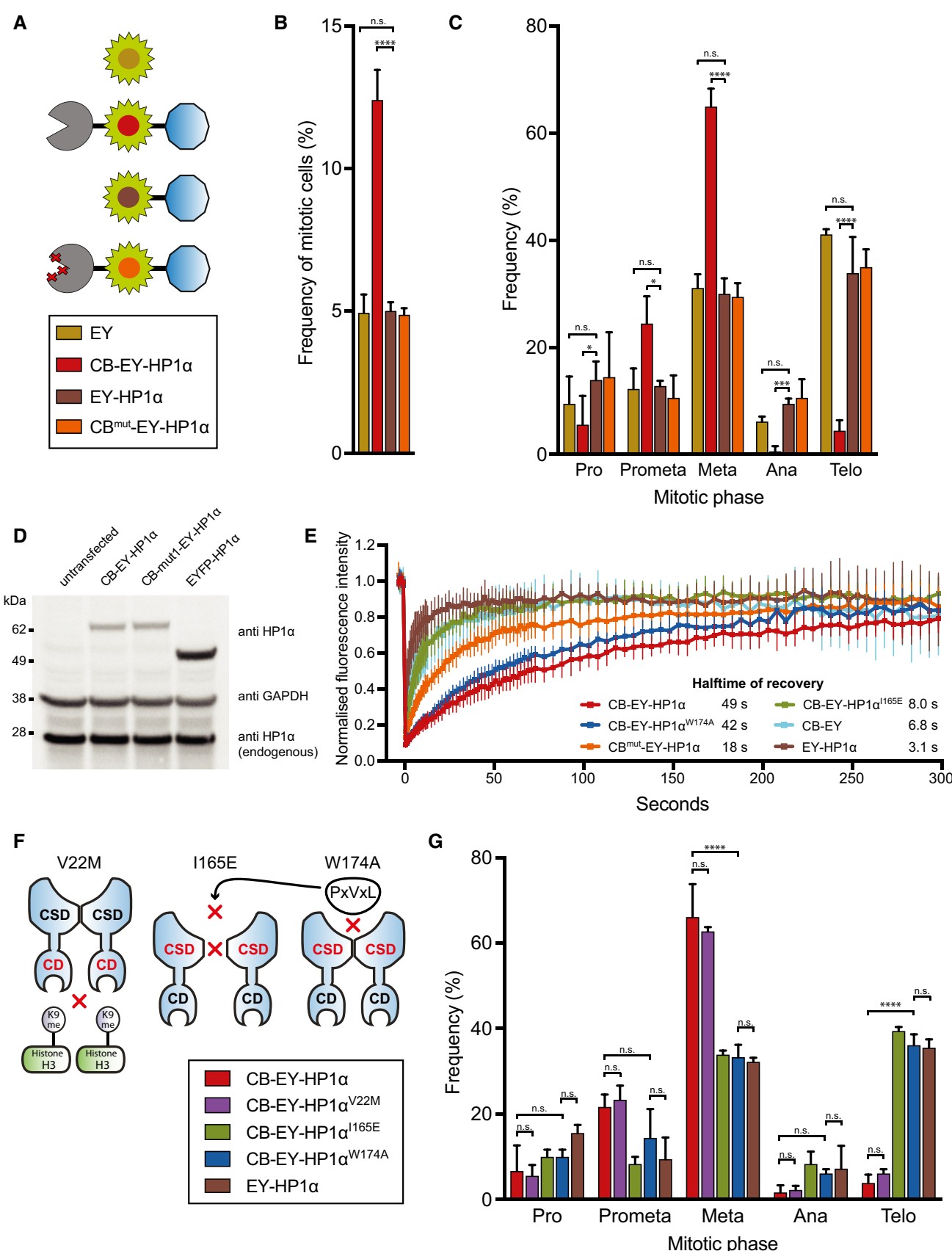

**Figure 1.**

fused protein, and suggests that CENP-B dimerisation is important for its stable binding to DNA.

Expression of CB-EY-HP1α[V22M] had no effect on the mitotic delay phenotype as expected, since it is tethered to centromeres via the CENP-B binding domain and does not need to recognise H3K9me3 for its localisation (Fig 1G). The single amino-acid substitutions I165E or W174A within the CSD completely eliminated the mitotic delay caused by HP1α tethering (Fig 1G). Importantly, in FRAP studies, the CSD mutant CB-EY-HP1α[W174A] showed similar dynamics to CB-EY-HP1α, with a mean half-time of recovery value of 42 s (Fig 1E). This may be because, like CB-EY-HP1α, this protein is able to dimerise. We used the CB-EY-HP1α[W174A] construct as a control in subsequent experiments, because it shows essentially identical expression, localisation and binding dynamics to wild-type CB-EY-HP1α, but does not cause the metaphase delay.

Together, these data suggest that the mitotic delay is caused by stable recruitment of a protein or proteins, presumably containing a PxVxL motif, that binds to the chimeric tethered HP1α at centromeres.

### The mitotic delay caused by centromeric tethering of HP1α is accompanied by a high frequency of segregation errors

Live cell imaging of HeLa cells confirmed the metaphase delay observed after centromeric tethering of wild-type HP1α. Imaging with differential interference contrast (DIC) optics allowed us to determine precisely the timing of nuclear envelope breakdown (NEB; Fig 2A—0 min) and anaphase onset (Fig 2A—84 min). Cells were scored separately depending on whether they expressed low (L), medium (M) or high (H) levels of chimeric protein (see Materials and Methods). We observed a median duration of 66 min (L), 120 min (M) and 111 min (H), respectively, from NEB until anaphase onset (Fig 2B). Strikingly, we could observe some cells remaining up to 38 h in mitosis before anaphase onset. Control cells expressing CB-EY-HP1α[W174A] showed a mitotic progression profile similar to untransfected cells. Similar results were obtained in U2OS osteosarcoma cells (Fig 2B).

Live cell imaging experiments revealed that HP1α tethering to centromeres resulted in a twofold to fourfold increase in the number of lagging chromosomes (Fig 2C). U2OS cells normally have a much higher baseline of lagging chromosomes, as previously reported (Kabeche & Compton, 2013). Strikingly, U2OS cells expressing CB-EY-HP1α at high levels exhibited lagging chromosomes in all cells (Fig 2C). As expected, the presence of lagging chromosomes was

correlated with an increased frequency of micronucleus formation (Fig 2D).

Thus, tethering HP1α to centromeres causes a strong phenotype with a mitotic delay accompanied by a high frequency of chromosome segregation errors.

### The mechanism of the delay suggests an involvement of the CPC

The mitotic delay induced by tethering HP1α to centromeres is due to activation of the spindle assembly checkpoint (SAC). Depletion of the essential SAC component mitotic arrest deficient 2 (Mad2) in HeLa cells expressing CB-EY-HP1α using published siRNA oligonucleotides (Gorbsky *et al*, 1998; Nitta *et al*, 2004; Fig 3A) resulted in a significant decrease in metaphase cells, and a concomitant increase in anaphase and telophase cells to levels comparable to those of control cells expressing CB-EY-HP1α[W174A] (Fig 3B).

To understand the reason for SAC activation, we used a cold-stable microtubule assay to determine whether HP1α tethering results in impaired microtubule attachment to kinetochores. Indeed, tethering of wild-type HP1α resulted in a reduced density of microtubules attached to kinetochores as well as kinetochores lacking any apparent microtubule attachments after cold treatment (Fig 3C). Measurement of the overall intensity of cold-stable microtubules revealed a clear contrast between cells expressing CB-EY-HP1α and control cells expressing CB-EY-HP1α[W174A] or untransfected cells (Fig 3D). Thus, tethering of CB-EY-HP1α decreases microtubule attachment to kinetochores.

We hypothesised that CB-EY-HP1α tethering to kinetochores might recruit the CPC, which is well known to regulate kinetochore–microtubule interactions (Cheeseman *et al*, 2006; DeLuca *et al*, 2006). We indeed observed an increased level of phosphorylated Dsn1, an Aurora B substrate (Welburn *et al*, 2010), at kinetochores in metaphase cells expressing CB-EY-HP1α compared to untransfected cells and cells expressing CB-EY-HP1α[W174A] (Fig 3E and F).

To further assess whether Aurora B activity causes the observed metaphase delay, we exposed CB-EY-HP1α-expressing cells to the Aurora B inhibitor ZM447439 (Fig 3G). Flow cytometry analysis revealed that this resulted in a decrease of the mitotic index to a level similar to that of control cells expressing CB-EY-HP1α[W174A] and untransfected cells.

Together, these results indicate that HP1α tethering using the CENP-B DBD results in a phenotype similar to that seen following CENP-B tethering of the core CPC protein INCENP, including metaphase delay, SAC activation, impaired microtubule attachments to

**Figure 2. Live cell imaging experiments show a robust mitotic delay and an increase in chromosome segregation defects caused by centromeric tethering of HP1α.**

A U2OS cell expressing high levels of CB-EY-HP1α from live cell imaging experiments. Arrows show lagging chromosome. Scale bar, 5 μm.

B Live cell analysis of the time from nuclear envelope breakdown (NEB) until anaphase onset of HeLa or U2OS cells expressing the indicated tethering construct or untransfected cells. Transient transfection resulted in low (L), medium (M) or high (H) protein expression levels (see Materials and Methods section). Crosses indicate cell death before anaphase onset and unfilled squares indicate end of the movie before anaphase onset. Graphs show median and interquartile range. Statistical significance was determined by the Kolmogorov–Smirnov test followed by the Benjamini–Hochberg multiple comparison test. ****$P < 0.0001$; n.s., not significant. Numbers of analysed cells are shown in Table EV1.

C, D Frequency of cells with lagging chromosomes or micronuclei observed in live cell experiments in cells expressing CB-EY-HP1α (red) or CB-EY-HP1α[W174A] (blue). Dark colour sections show frequencies of cells with lagging chromosomes or micronuclei, and pale colour sections show frequencies of cells without lagging chromosomes or micronuclei. Statistical significance was determined by Fisher's exact test followed by the Benjamini–Hochberg multiple comparison test. *$P < 0.05$; **$P < 0.01$; ***$P < 0.001$; ****$P < 0.0001$; n.s., not significant.

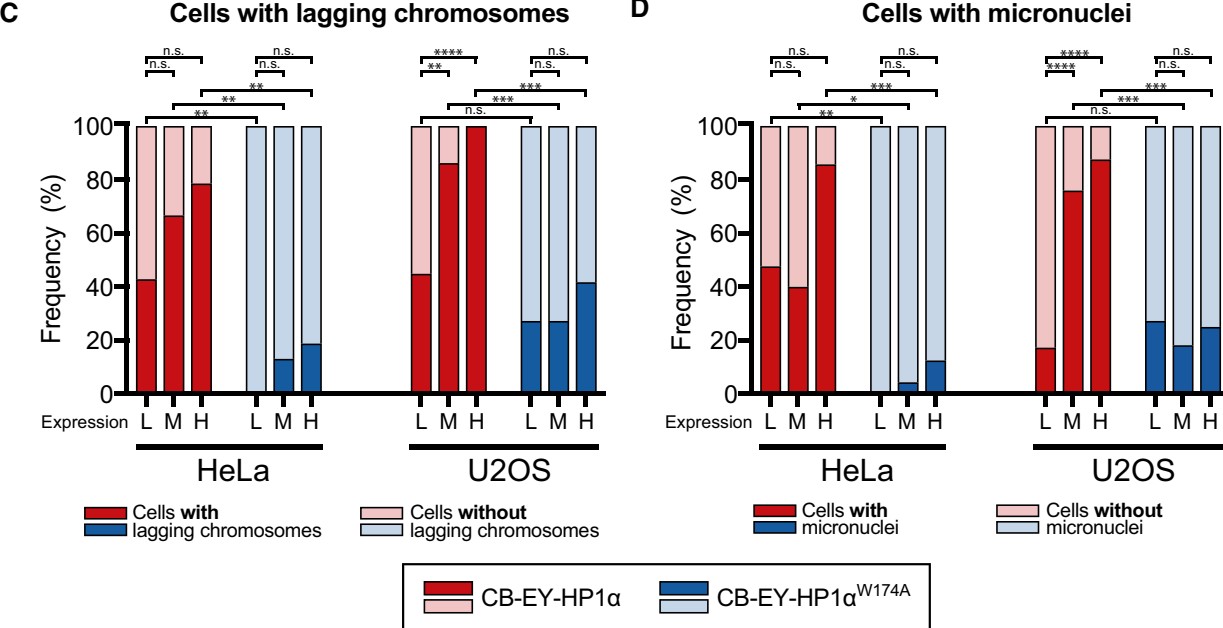

**Figure 2.**

kinetochores, increased Aurora B activity and ZM447439 sensitivity (Liu *et al*, 2009).

## Tethering of HP1α to centromeres leads to abnormal centromeric retention of Aurora B in telophase

Cells expressing CB-EY-HP1α showed an abnormal retention of Aurora B at centromeres in anaphase and even after mitotic exit, when chromosomes start to decondense (Fig EV2A and B). This strong co-localisation of Aurora B with CB-EY-HP1α at centromeres was clearly seen in telophase, where only a fraction of the kinase exhibited its physiological localisation to the midbody region, with a significant portion of the kinase remaining at centromeres (Fig 4A1). This centromeric Aurora B was not observed in control telophase cells expressing CB-EY-HP1α$^{W174A}$ (Fig 4A2).

CB$^{mut}$-EY-HP1α was much less efficient at retaining Aurora B at centromeres during telophase. Instead, we detected EYFP signal at the midbody (Fig 4A3). Thus, with this construct, the CPC was dominant and determined the localisation of the CB$^{mut}$-EY-HP1α fusion construct. Recruitment of HP1α to the midbody region was also observed after expression of untethered EY-HP1α (Fig 4A4) and has been previously reported by others (Hayakawa *et al*, 2003).

Together, these results reveal the existence of unexpectedly robust interactions between HP1α and the CPC in late mitosis. Thus, tethered HP1α can retain the CPC at centromeres, and the CPC can recruit untethered or weakly tethered HP1α to the midbody.

## Constitutive retention of HP1-bound CPC results in H3S10 phosphorylation in $G_1$

Remarkably, interphase cells in $G_1$ phase (cyclin A2 negative) expressing CB-EY-HP1α exhibited a robust centromeric H3S10ph signal that was not detectable in cells expressing CB-EY-HP1α$^{W174A}$ (Fig 4B). Automated image analysis detected H3S10ph foci in 54% of CB-EY-HP1α-expressing $G_1$ cells 24 h after transfection (Fig 4C). In contrast, the same detection parameters revealed few, if any, H3S10ph-positive $G_1$ cells in cultures expressing CB-EY-HP1α$^{W174A}$, CB$^{mut}$-EY-HP1α, EY-HP1α, or in untransfected cells, none of which retained Aurora B at telophase centromeres (Fig 4A). Comparable results were obtained in U2OS cells (Fig 4C).

This H3S10ph signal persisted throughout interphase. Using bead loading of fluorescently labelled antigen-binding fragments (Fabs; Hayashi-Takanaka *et al*, 2009), we could observe H3S10ph-labelled foci for over 8 h after mitotic exit (Fig EV3A, Movie EV1) and even track H3S10ph in favourable cells across an entire cell cycle between two consecutive mitoses (Fig EV3B, Movie EV2). The persistence of this signal required continued Aurora B activity. The H3S10ph signal in interphase cells disappeared after the addition of 0.5 μM ZM447439, despite the continued persistence of Aurora B in CB-EY-HP1α foci (Fig EV2C and D). This dose of drug has little apparent effect on H3S10ph in mitosis (Fig EV2C). This is consistent with an altered kinase/phosphatase balance in interphase and mitotic cells.

Thus, stably tethered HP1α is able to localise a functional CPC even in $G_1$ cells, a stage of the cell cycle at which the CPC is normally inactive.

## Characteristic labelling of endogenous H3S10ph foci in $G_2$ cells at the CDK1 arrest point

H3S10ph, the most widely studied read-out of Aurora B activity, has been known for many years to be associated with mitotic chromosome condensation (Gurley *et al*, 1978). Development of a specific antibody recognising this modification led to the recognition that the mark is first abundant during $G_2$ phase (Hendzel *et al*, 1997), although it can also be detected at much lower levels at promoters during gene activation (Nowak & Corces, 2004). Given the ability of tethered HP1α to activate the CPC during interphase, we decided to examine the relationship between HP1 isoforms and H3S10ph during $G_2$ phase. Indeed, co-staining readily revealed that H3S10ph foci in $G_2$ cells co-localised with untethered EY-HP1α (Fig 5A).

H3S10ph shows a spectrum of different staining patterns in $G_2$ cells (Fig 5B). These include nuclei with a few isolated foci, nuclei in which the foci are larger and more abundant, and nuclei that show a general more diffuse labelling throughout the chromatin. In order to unambiguously determine whether these different patterns reflect a temporal progression as cells enter prophase, we used chemical genetics to create HeLa cells whose cell cycle is driven by a CDK1-as allele that is inhibited by the ATP analogue 1NM-PP1 (Fig 5C). 1NM-PP1 arrests these cells late in $G_2$ and cells enter

---

**Figure 3. The mechanism of the mitotic delay after HP1α tethering suggests an involvement of the CPC.**

A   Immunoblot of HeLa cell lysates transfected with the indicated siRNA shows depletion of Mad2 protein in cells transfected with siRNA targeting Mad2 mRNA. α-tubulin was a loading control.

B   Frequency of mitotic phases in HeLa cells 24 h after transfection. Cells were co-transfected with control siRNA (solid bars) or with Mad2 siRNA (striped bars). Graphs show the mean and SD of three independent experiments (*n* = 60 mitotic cells per experiment). Statistical significance was determined by Fisher's exact test followed by the Benjamini–Hochberg multiple comparison test. ***P < 0.001; ****P < 0.0001; n.s., not significant.

C   HeLa cells expressing the indicated HP1α fusion proteins or untransfected cells after cold treatment. Cells were stained with Hoechst 33342 and immunostained for α-tubulin and CENP-C. Scale bar, 5 μm. Merge shows a maximum intensity projection of five *z*-planes. Zoom shows either α-Tubulin and CENP-C (i, ii), tethering construct and CENP-C (iii and iv), or the tethering construct only (v, vi).

D   Quantification of the microtubule intensity of HeLa cells after cold treatment. Graphs show mean and SD of normalised values from three independent experiments (*n* = 33 of untransfected cells, *n* = 36 of transfected cells). Statistical significance was determined by the Kolmogorov–Smirnov test.

E   HeLa cells stained with Hoechst 33342 and immunostained for phosphorylated Dsn1 and Hec1 after pre-extraction. Scale bar, 5 μm.

F   Quantification of the mean Dsn1ph value per kinetochore in metaphase cells. Graphs show median and interquartile range of three independent experiments. Individual kinetochores of 60 (untransfected and CB-EY-HP1α expressing) or 58 (CB-EY-HP1αW174A expressing) cells were analysed and compared. Statistical significance was determined by the Kolmogorov–Smirnov test. ****P < 0.0001.

G   HeLa cells were treated with DMSO or 3 μM ZM447439 24 h after transfection. Mitotic indices were determined by flow cytometry after cells were stained with Hoechst 33342 and immunostained for MPM2. At least 40,000 cells were analysed per condition and experiment. Graphs show the mean and SD of three independent experiments.

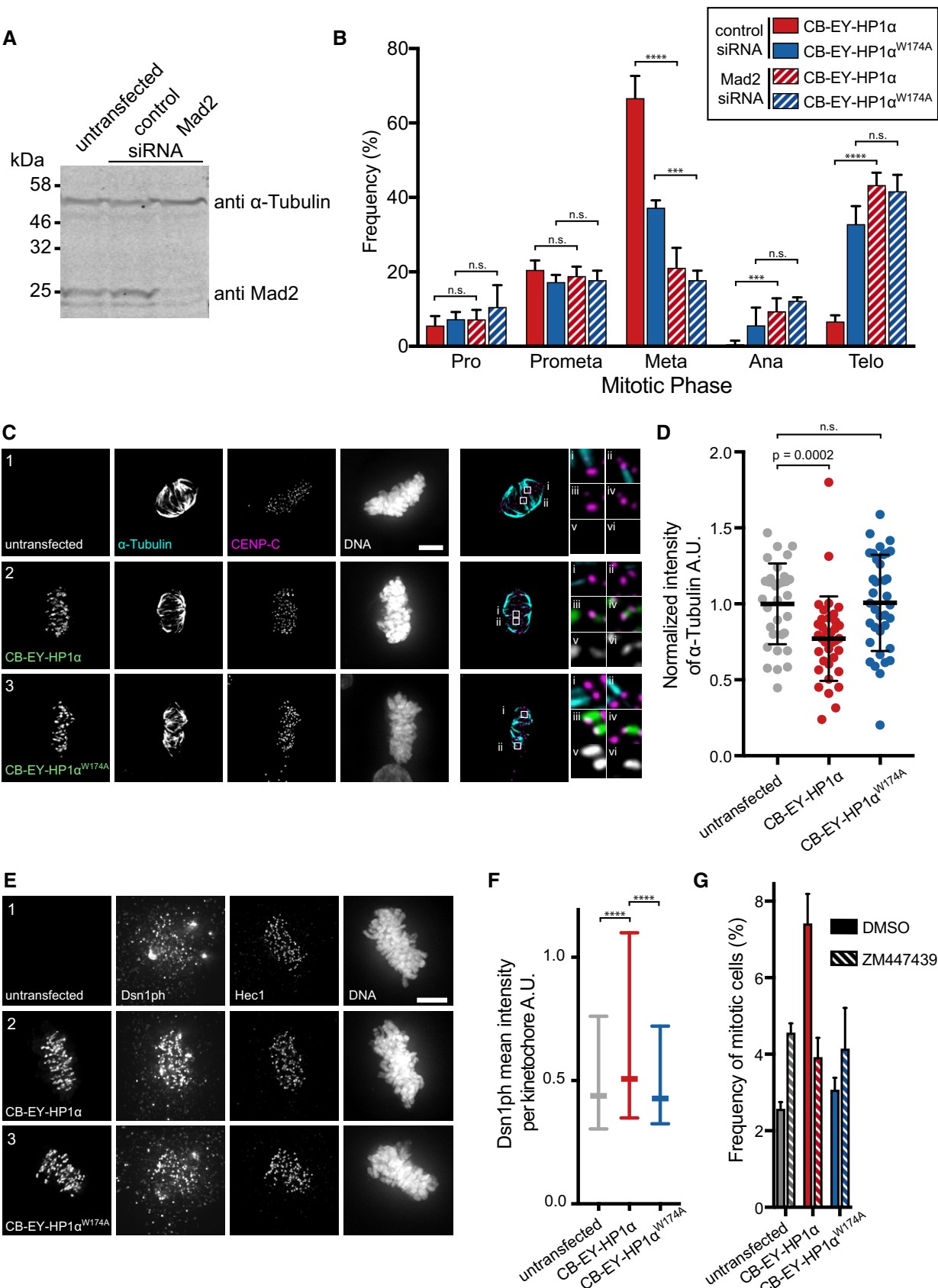

**Figure 3.**

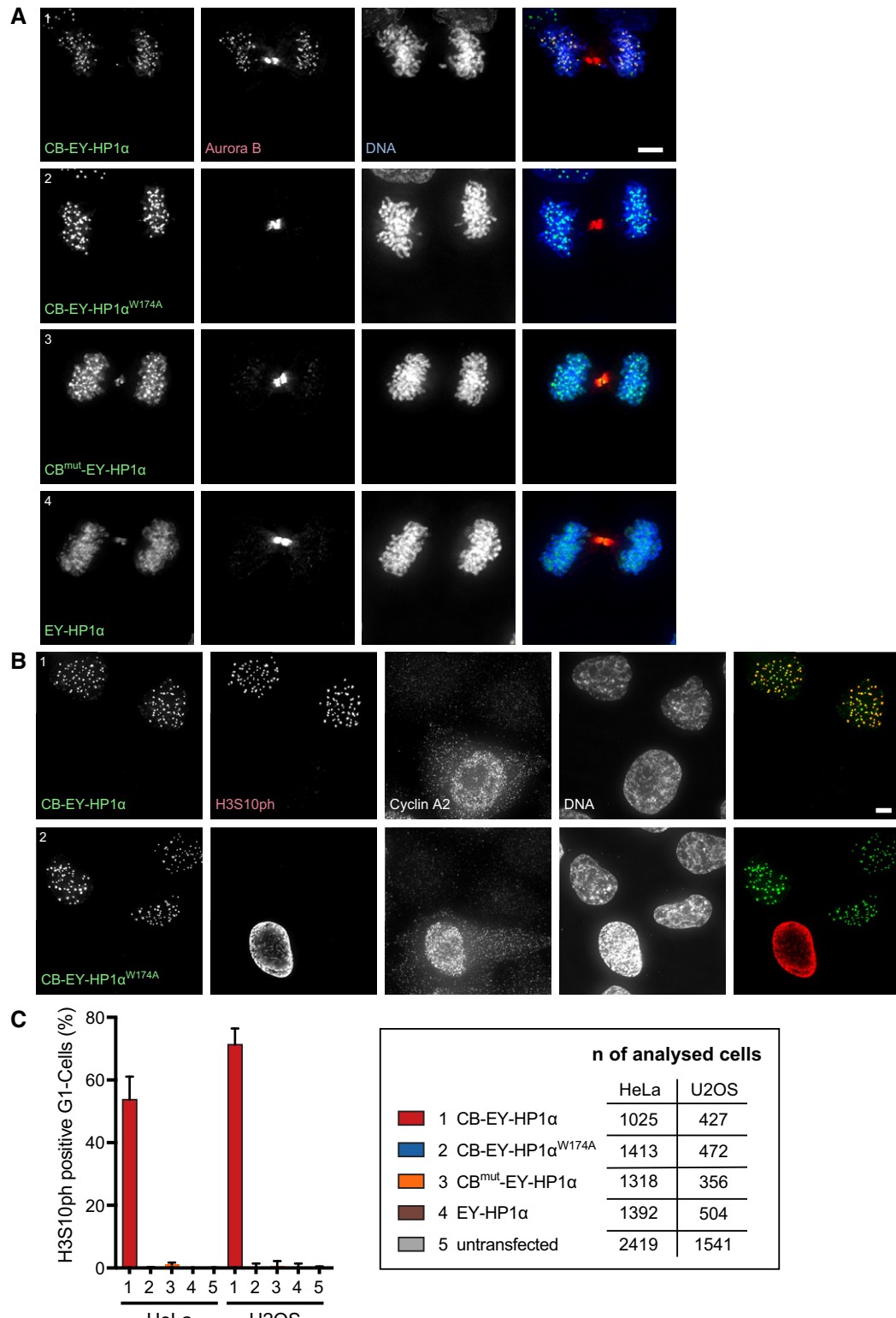

**Figure 4.  Constitutive retention of HP1-bound CPC at centromeres results in H3S10 phosphorylation in G₁.**

A   HeLa cells stained with Hoechst 33342 and immunostained for Aurora B. Scale bar, 5 µm.

B   HeLa cells stained with Hoechst 33342 and immunostained for histone H3 Serine10 phosphorylation (H3S10ph) and cyclin A2 to identify cells in G₁ (cyclin A2-negative cells). Scale bar, 5 µm.

C   Quantification of histone H3S10ph-positive G₁ cells 24 h after transfection of the indicated HP1α fusion constructs or in untransfected cells. Graphs show mean and SD of three independent experiments. Total numbers of analysed cells that were transfected and negative for cyclin A2 staining are shown.

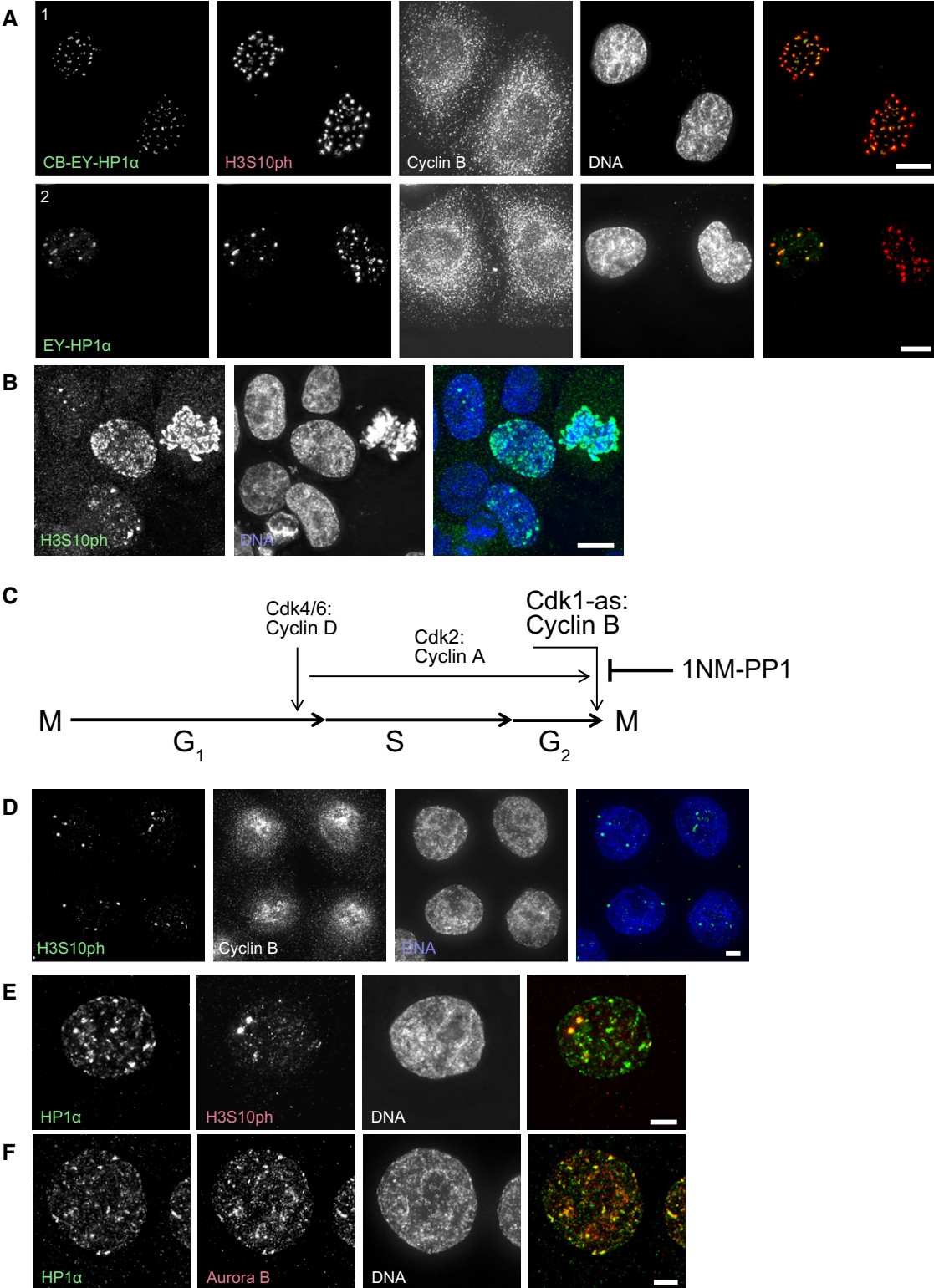

**Figure 5. Characteristic labelling of endogenous H3S10ph foci in G₂ cells at the CDK1 arrest point.**

A    HeLa cells stained with Hoechst 33342 and immunostained for H3S10ph and cyclin B. Scale bar, 10 μm.
B    Unsynchronised HeLa CDK1-as cells stained with Hoechst 33342 and immunostained for H3S10ph. Scale bar, 10 μm.
C    Schematic representation of the effect of 1NM-PP1 treatment on cell cycle progression in cells with an analogue sensitive CDK1 kinase (CDK1-as).
D–F  HeLa CDK1-as cells treated with 10 μM 1NM-PP1 for 20 h, stained after pre-extraction with Hoechst 33342 and immunostained for either H3S10ph and cyclin B (D), HP1α and H3S10ph (E), or HP1α and Aurora B (F). Scale bars, 5 μm.

mitosis in under 60 min after removal of the drug. This allowed us to study the emergence of H3S10ph foci in $G_2$ cells with high temporal precision.

Examination of cultures after 20 h of 1NM-PP1 incubation revealed a remarkable uniformity of the H3S10ph staining, with almost every cell having 3–6 bright H3S10ph foci (Fig 5D). When labelled Fab fragments recognising H3S10ph were introduced into living cells in the presence of 1NM-PP1, the foci were found to be highly stable and did not change over a period of 12 h (Fig EV4B, Movie EV3).

These H3S10ph foci co-localise with clusters of endogenous HP1α (Fig 5E), and endogenous HP1α co-localises with Aurora B kinase (Fig 5F). As was the case for the CPC recruited to centromeres in $G_1$ phase by tethered HP1α, treatment with 0.5 μM of the Aurora B inhibitor ZM447439 completely abolished the H3S10ph staining in the synchronised culture, although Aurora B still co-localised with EY-HP1α foci (Fig EV4A). This co-localisation between Aurora B and EY-HP1α was also observed in unsynchronised cells, where treatment with 0.5 μM ZM447439 abolished the H3S10ph signal (which remained readily detectable in mitotic cells).

Thus, localised H3S10 phosphorylation begins at HP1 foci during $G_2$ prior to CDK1-cyclin B activation.

### H3S10 phosphorylation precedes H3T3 phosphorylation in $G_2$

It is now widely accepted that survivin binding to H3T3ph has an important role in localising of the CPC to centromeres during mitosis. We therefore investigated whether this modification was involved in targeting the CPC to its sites of action during $G_2$ phase.

No H3T3ph signal was detectable in the culture after synchronisation of CDK1-as cells with 1NM-PP1, even though almost every cell showed three to six prominent H3S10ph foci (Fig 6A). To exclude the possibility that the CDK1 inhibition was interfering with Haspin activity in these synchronised $G_2$ cells, we also analysed unsynchronised cells (−1NM-PP1). Again, the H3S10ph foci appeared before H3T3 phosphorylation was detected, which typically occurred when the nucleus exhibited general chromatin staining for H3S10ph. In a further control, we stained for H3T3ph and H3S10ph in wild-type HeLa cells (Fig 6B). This yielded the same result: strong H3S10ph foci were visible in H3T3ph-negative cells, and H3T3ph was only visible in cells with a strongly H3S10ph-positive nucleus.

To further resolve the temporal relationship of H3S10ph and H3T3ph in cycling cells, we bead-loaded HeLa cells with Alexa488-labelled Fab fragments against H3S10ph and CF640R-labelled Fab fragments against H3T3ph. This allowed a very clear temporal resolution of the formation of the two marks in living cells (Fig 6C, Movie EV4). This analysis demonstrated that H3S10ph foci are established at centromeres long before H3T3ph emerges. Interestingly, H3T3ph disappears rapidly after anaphase onset, whereas H3S10ph persists for a longer time (Fig 6C—10.8 h).

### Loss of HP1α and HP1γ abolishes H3S10ph foci in $G_2$ cells

In the light of our HP1 tethering experiments demonstrating the strong interaction between HP1α and the CPC and the clear co-localisation between H3S10ph foci and clusters of HP1α, we wished to determine whether HP1α is required for the clustering and activation of Aurora B in $G_2$ cells.

To probe the requirement for HP1 isoforms in H3S10ph focus formation in $G_2$ cells, we created single and double knockouts of HP1α, HP1β and HP1γ in HeLa cells (Fig EV5A). We used the CDK1 inhibitor RO-3306 to synchronise cells in $G_2$, because the HP1 knockouts were performed in cells with a wild-type CDK1. We stained for cyclin B to assess the cell cycle stage of these cells more precisely. As in previous experiments, robust H3S10ph foci were visible in wild-type $G_2$ cells (Fig 7A). In HP1α KO cells, H3S10ph foci were still detectable in cyclin B-positive $G_2$ cells, but with a reduced frequency (93% in WT vs. 67% in the HP1α KO).

Testing of the various HP1 mutants revealed that only the double knockout (DKO) of HP1α + HP1γ resulted in a loss of nearly all discrete H3S10ph foci in $G_2$ cells. These nuclei exhibited a diffuse background labelling for H3S10ph (Fig 7A) as well as a diffuse distribution of Aurora B kinase (Fig 7B). Expression of EY-HP1α in these DKO cells could rescue this phenotype, and discrete H3S10ph foci co-localised with the transfected EY-HP1α (Fig 7C).

In very rare (< 5%) cases where H3S10ph clusters were observed (Fig 7A), those DKO cells exhibited much stronger cyclin B staining, suggesting that they were nearing the $G_2/M$ transition. Indeed, live cell imaging with Fab fragments revealed that H3S10ph clusters are present long before mitosis in wild-type cells, whereas weak H3S10ph clusters appeared usually only four frames (24 min) before NEB in HP1α + HP1γ DKO cells (Fig EV5B, Movie EV5).

Co-staining for H3T3ph and H3S10ph in fixed cells and live cell imaging of these chromatin marks with Fab fragments in the HP1α + HP1γ DKO revealed that the time difference between the robust appearance of these chromatin marks is apparently much smaller in these cells (Fig EV5C and D, and Movie EV6). Importantly, entry of the HP1α + HP1γ DKO cells into mitosis was ultimately accompanied by a strong wave of H3S10 phosphorylation. This is consistent with there being several redundant pathways for targeting the CPC to chromatin in mitosis.

## Discussion

The CPC is an integral part of a centromeric signalling network that regulates chromosome segregation in mitosis. CPC functions during mitosis are widely studied, but much less is known about CPC activation in $G_2$ phase, when Aurora B activity is first detected. These early $G_2$ events may be important: it has been reported that the timing of Aurora B activation (detected as phosphorylation of H3S10) correlates with the accuracy of chromosome segregation (Hayashi-Takanaka et al, 2009). H3S10ph foci appear later in $G_2$ in cells that suffer a high frequency of chromosome missegregation in the subsequent mitosis.

We have focused on interactions between HP1α and the CPC in mitosis and interphase. Our studies artificially tethering HP1α to centromeres revealed a robust interaction between this protein and the CPC. Indeed, HP1α tethered in kinetochore-proximal chromatin by the CENP-B DNA-binding domain produces phenotypes (including an increased mitotic index due to SAC activation by destabilised kinetochore–microtubule interactions) that resemble the effects produced by directly tethering the core CPC subunit INCENP to this location using a similar CENP-B DBD fusion (Liu et al, 2009). Overall, our findings are consistent with a recent suggestion that HP1α

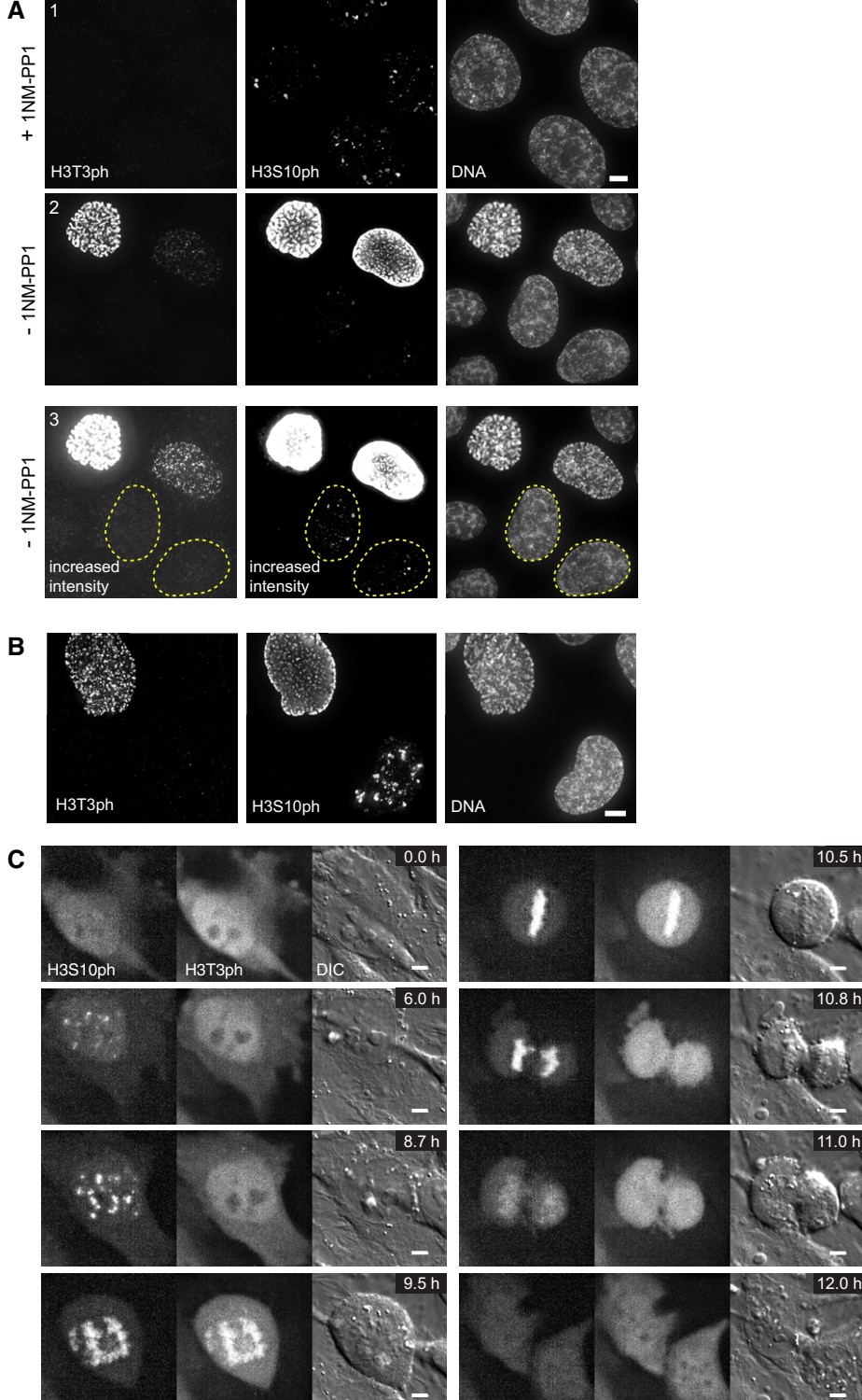

**Figure 6. H3S10 phosphorylation precedes H3T3 phosphorylation in G₂.**

A   HeLa CDK1-as cells treated with either 10 μM 1NM-PP1 (+1NM-PP1) or DMSO (−1NM-PP1) for 20 h, stained with Hoechst 33342 and immunostained for H3S10ph and H3T3ph. Panel A3 is the same as panel A2 but with increased intensities. Outlined nuclei highlight the stage where H3S10ph is already present while H3T3ph is still absent. Scale bar, 5 μm.

B   Wild-type HeLa cells stained with Hoechst 33342 and immunostained for H3S10ph and H3T3ph. Scale bar, 5 μm.

C   Stills of a live cell imaging movie using Alexa488-labelled Fabs against H3S10ph and CF640R-labelled Fabs against H3T3ph in HeLa cells. Images were acquired every 10 min with five *z* sections every 1.2 μm. Scale bars, 5 μm.

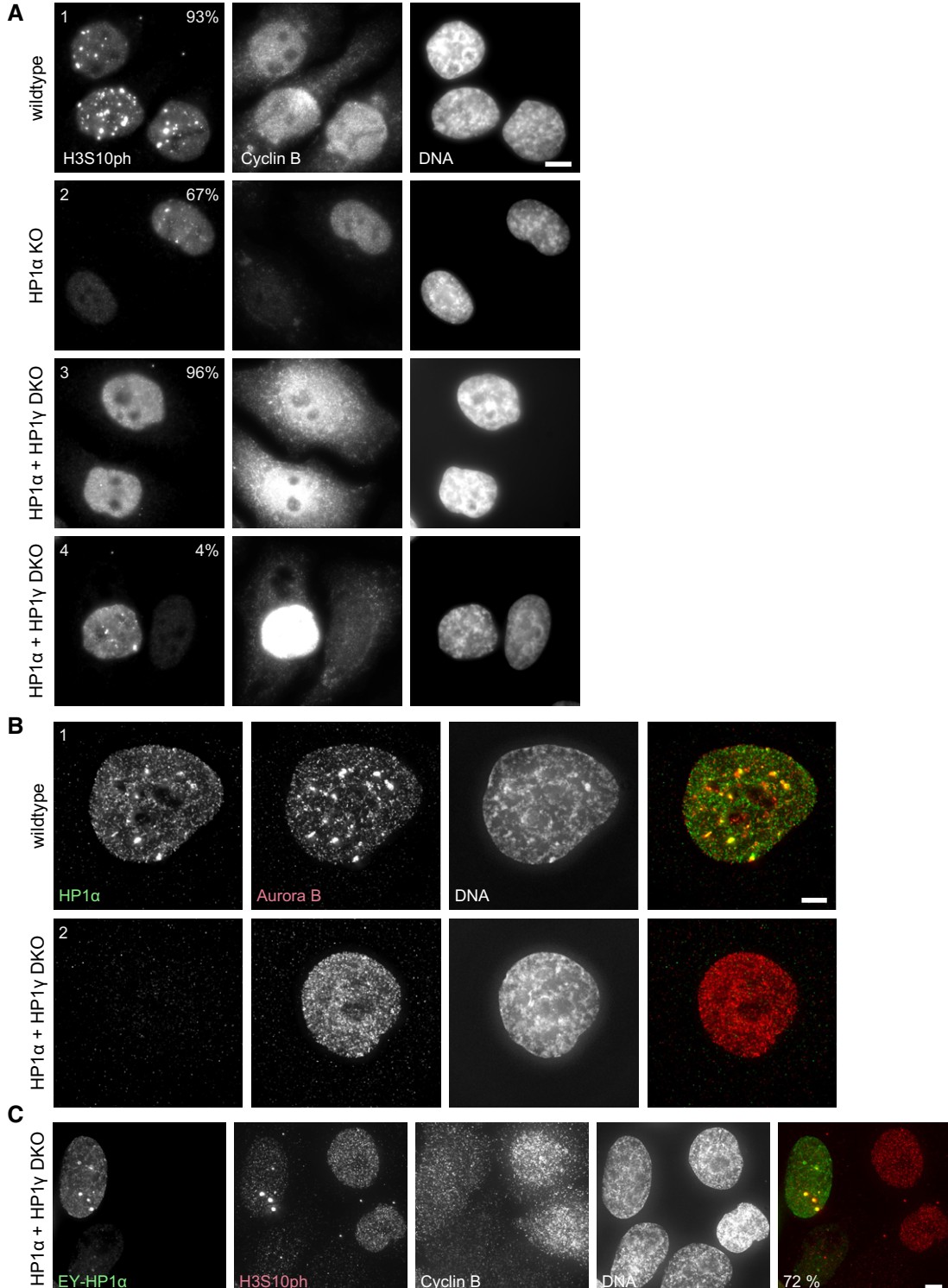

**Figure 7. Loss of HP1α and HP1γ abolishes H3S10ph foci in G₂ cells.**

A   HeLa wild-type (1), HP1α KO (2) or HP1α and HP1γ double KO (DKO) (3, 4) cells were synchronised with 9 μM RO-3306 for 18 h. Cells were stained with Hoechst 33342 and immunostained for H3S10ph and cyclin B. No deconvolution was performed to preserve the true appearance of the cyclin B staining. The per cent of cells with the phenotype shown is indicated (*n* = 100). Scale bar, 10 μm.

B   HeLa wild-type (1), or HP1α and HP1γ double KO (2) cells synchronised with 9 μM RO-3306 for 18 h. Cells were stained with Hoechst 33342 and immunostained for HP1α and Aurora B. All interphase HP1α and HP1γ double KO cells (2) exhibited a diffuse localisation of Aurora B. Scale bar, 5 μm.

C   HeLa HP1α and HP1γ double KO cells transfected with EY-HP1α and 24 h after transfection synchronised with 9 μM RO-3306 for 12 h. Cells were stained with Hoechst 33342 and immunostained for H3S10ph and cyclin B. The per cent of EY-HP1α expressing cells with H3S10ph clusters is indicated (*n* = 100). Scale bar, 5 μm.

can function as an additional subunit of the CPC (Nozawa *et al*, 2010; Abe *et al*, 2016).

Tethered HP1α can trap the CPC at centromeres in anaphase, preventing it from undergoing its normal transfer to the central spindle and cell cortex. This trapped CPC remains clustered at centromeres during $G_1$ and the kinase retains its catalytic activity, producing persistent foci of H3S10ph that remain throughout interphase. These foci vanish if Aurora B is inhibited with low doses of ZM447439 that have no obvious effect on H3S10ph during mitosis. The CPC is well known to be activated by clustering (Sessa *et al*, 2005; Kelly *et al*, 2007; Fuller *et al*, 2008; Tseng *et al*, 2010; Wang *et al*, 2011). Our data reveal that stably targeted HP1α (CB-EY-HP1α) can recruit CPC clusters in which the kinase activity is maintained even in the presence of interphase levels of competing phosphatase activity.

Aurora B activity is normally down-regulated during mitotic exit by Cdh1-dependent proteasomal degradation (Nguyen *et al*, 2005; Stewart & Fang, 2005) and increased phosphatase activity (Murnion *et al*, 2001; Nguyen *et al*, 2005; Stewart & Fang, 2005; Vagnarelli *et al*, 2011; Wurzenberger & Gerlich, 2011; Lee *et al*, 2016). However, Aurora B tethered in clusters at centromeres by CB-EY-HP1α is stable throughout interphase: its degradation during mitotic exit may require transfer of the CPC away from centromeres. This is consistent with the observation that Cdh1 can exhibit differential activity against different pools of Aurora B in $G_1$ (Floyd *et al*, 2013).

One dominant paradigm for CPC activation in mitosis is that Haspin kinase phosphorylation of H3T3 creates a binding site for the BIR domain of survivin, and this clusters the CPC at centromeres. Our experiments using Fab fragments to monitor the dynamic behaviour of histone modifications in living cells reveal that H3S10ph appears in $G_2$ long before H3T3ph, as previously suggested using fixed cells (Polioudaki *et al*, 2004). The H3T3ph signal appears only as the H3S10ph signal starts to spread across the chromatin. It is therefore tempting to speculate that as chromatin becomes generally phosphorylated on H3S10, the association of HP1 with the chromatin is weakened by the methyl-phos switch (Fischle *et al*, 2005; Hirota *et al*, 2005), and H3T3ph acts as a new signal to re-concentrate the CPC at centromeres. The reason for this complex dual control of CPC localisation is unknown.

Just as tethered CB-EY-HP1α can induce the formation of active CPC clusters in $G_1$ cells, our subsequent results suggest that the first activation of the CPC during $G_2$ is mediated by interactions with endogenous HP1α. Our experiments with CDK1-as cells arrested by 1NM-PP1 reveal that this early CPC does not require CDK1 activity for its activation—these cells exhibit three to six bright foci of H3S10ph that co-localise with HP1α. Thus, HP1-induced clustering appears to promote CPC activation, even under conditions where competing phosphatases, which are inactivated by CDK1 and other mitotic kinases, retain their interphase level of activity (Wurzenberger & Gerlich, 2011; Heim *et al*, 2017). The functional consequences of the early CPC activation at HP1 foci are not known; however, a recent study reports that knockout of mouse HP1α is associated with increased merotelic chromosome attachments during mitosis (Bosch-Presegué *et al*, 2017), consistent with impaired CPC activity (Gassmann *et al*, 2004; Cimini *et al*, 2006).

Both HP1α and HP1γ appear to be involved in clustering and activating the CPC, as single knockouts of either protein still exhibit foci of H3S10ph during $G_2$. Only the double knockout shows diffuse

labelling of the chromatin. In the absence of HP1α + HP1γ, foci of Aurora B activity appear only just prior to entry into prophase, consistent with the timing of the appearance of H3T3ph label. It is possible that the late $G_2$ activation of Aurora B in the absence of HP1α and HP1γ may involve clustering of the CPC at sites of Haspin activity near centromeres.

Live cell imaging using labelled Fab fragments recognising H3S10ph and H3T3ph in the same cells revealed that H3T3ph disappears rapidly after anaphase onset as previously reported (Dai *et al*, 2005; Kelly *et al*, 2010; Qian *et al*, 2011), but interestingly, H3S10ph persists for somewhat longer. We speculate that this might allow the proper transition of the CPC to the central spindle, with survivin no longer binding to H3T3ph and HP1 binding to H3K9me3 continuing to be inhibited. This is in line with our tethering experiments revealing that HP1α's chromatin binding is regulated rather than its attachment to the CPC. This could potentially explain why H3S10ph is so broadly conserved during mitosis: the methyl/phos switch releasing HP1 from chromatin might help to ensure subsequent CPC mobility. This could promote the shift from HP1-directed interphase CPC localisation to the mitotic localisation directed by H3T3ph and H2AT120ph. Indeed, ZM447439 treatment, which leads to diminished H3S10ph and H3T3ph in mitosis, results in decreased centromeric localisation of HP1 and INCENP, accompanied by an increased HP1-dependent chromosome arm localisation of INCENP (Nozawa *et al*, 2010).

Two previous studies reported that Sgo1 localisation in interphase cells depends on HP1 (Perera & Taylor, 2010; Kang *et al*, 2011). HP1-dependent clustering of Sgo1 at centromeres in early $G_2$ was followed by diffuse staining for the remainder of $G_2$, and then a switch to Bub1/H2AT120ph-dependent binding to kinetochores at mitotic entry (Perera & Taylor, 2010). Those studies, together with the results reported here reveal that in addition to participating in heterochromatin formation via phase separation (Larson *et al*, 2017; Strom *et al*, 2017), HP1 also plays important roles in the regulation of components that regulate centromeric cohesion (Sgo1) and kinetochore function (the CPC) prior to and during mitotic entry. HP1-induced CPC clustering appears to be an effective way of promoting the activation of Aurora B kinase.

## Materials and Methods

### Constructs

The DNA fragment corresponding to the DNA-binding domain (DBD) of human CENP-B (aa 1–159) was optimised for human cell line expression and synthesised by Geneart (Thermo Fisher Scientific) and cloned into the NheI and AgeI restrictions sites of the pYIP-EYFP vector, containing attL and attR sites for Gateway cloning (Molina *et al*, 2016). HP1α was amplified from a custom-made cDNA library from HeLa cells using following oligonucleotides as primers: HP1α-Fwd (5′-CACCATGGGAAAGAAAACCAAGCGGA CAGC-3′) and HP1α-Rev (5′-GCTCTTTGCTGTTTCTTTCTCTTTG TTTTCC-3′). The PCR product was cloned into the pENTR plasmid and Gateway cloning was performed according to the manufacturer' instructions (Thermo Fisher Scientific) to generate a plasmid coding for the fusion construct CENP-B$^{DBD}$-EYFP-HP1α, which is expressed under a CMV promoter. This plasmid was lacking a stop codon directly after the HP1α sequence to allow the potential fusion of

additional proteins. However, for the final plasmid a stop codon was introduced after the codon coding for S191 of the HP1α sequence via site-directed mutagenesis using the QuikChange II kit (Stratagene). Site-directed mutagenesis was also used to generate the three HP1 mutants CENP-B$^{DBD}$-EYFP-HP1α$^{V22M}$, CENP-B$^{DBD}$-EYFP-HP1α$^{I165E}$, CENP-B$^{DBD}$-EYFP-HP1α$^{W174A}$. To generate CENP-B$^{DBD-mut}$-EYFP-HP1α, the DNA fragment corresponding to the DBD of human CENP-B (1–159aa), but coding for the substitutions S40A, N120A, R125A, was synthesised by Geneart and cloned into the NheI and AgeI restriction sites of the pYIP CENP-B$^{DBD}$-EYFP-HP1α plasmid, replacing the wild-type CENP-B$^{DBD}$. The pYIP-EYFP-HP1α plasmid was generated by digestion of the pYIP CENP-B$^{DBD}$-EYFP-HP1α plasmid with ClaI and AgeI, removing the CENP-B$^{DBD}$ sequence between those restriction sites.

### Generation of the CDK1-as cell line

HeLa analogue sensitive CDK1 (CDK1-as) cells were created using the MKF1 HeLa cell line previously described (Klebig *et al*, 2009) based on the system previously used by (Hochegger *et al*, 2007). In brief, *Xenopus* CDK1as cDNA linked to a puromycin resistance gene by a T2A peptide was transfected into the MKF1 cell line. Clones resistant to puromycin with adequate levels of XsCDK1asT2Apuro were selected by Western blot analysis. Endogenous *human* CDK1 was inactivated by transient transfection of plasmids encoding hCas9 (Addgene: ID41815) and hCdk1 gRNA against the following targeting sequence ATTTCCCGAATTGCAGTACTAGG. hCdk1 gRNA was cloned into the following plasmid (Addgene:ID41824). The indicated Addgene plasmids were a gift from George Church (Mali *et al*, 2013). Final clones were tested with 1NM-PP1 for block in $G_2$ and subsequent entry in mitosis following 1NM-PP1 washout.

### Generation of HP1 KO cell lines

pTORA14HA3, pTORA14HB7 or pTORA14HG1 plasmids to generate HP1α, HP1β or HP1γ KO, respectively, were constructed through inserting the double-strand oligo DNA fragment in the AgeI site, blunted by mung bean nuclease of the plasmid vector (pTORA14) to produce Cas9-GFP from the CMV promoter. The guide RNA target sequences in the DNA fragment were 5′-CGCGCCTGTCTAGCACCTT-3′, 5′-CCCTCTGATTTATCTGTCT-3′ and 5′-GACAAATTCTTCAGGCTCT-3′, respectively, for the three plasmids. HeLa cells transfected with plasmids pTORA14HA3, pTORA14HB7 or pTORA14HG1 were sorted based on GFP intensity, using FACSAria II (BD Biosciences). Genomic DNA was extracted from sorted cell lines, and loci targeted by guide RNAs were amplified and sequenced using primers (5′-TTTGAGACTCAAGAGCAGGG-3′ and 5′-AACGTAAGCTCCACAAGCGG-3′ for HP1α KO; 5′-ATTTGCCTTTGAAGGAAGCC-3′ and 5′-TTTTCCATTTACTGCTCCTG-3′ for HP1β KO; 5′-ATTTTGGTGGTGGGTTGTAAG-3′ and 5′-TAGACCTCAAATGAGACACC-3′ for HP1γ KO). HP1α and HP1γ double KO or HP1β and HP1γ double KO were generated from HP1γ KO cell.

### Cell culture, RNA interference and drug treatment

HeLa and U2OS cells were cultured in DMEM supplemented with 10% foetal bovine serum (FBS), 100 U/ml penicillin, and 100 μg/ml streptomycin at 37°C and 5% $CO_2$ in air.

Transient transfection for indirect immunofluorescence experiments was performed using jetPRIME (Polyplus Transfection) according to the manufacturer's instructions. In short, cells were seeded the day prior transfection in 12-well plates on polylysine-coated coverslips. Plasmid DNA (125–500 ng) and, where indicated, siRNA oligonucleotides (final concentration of 50 nM) were added to 100 μl of jetPRIME buffer. In addition, 125 ng of salmon sperm DNA (UltraPure Salmon Sperm DNA Solution, Thermo Fisher Scientific) or 40 nM of the 21-mer oligonucleotide CGUACGCGGAAUACUUCGAdTdT (Elbashir *et al*, 2001) was added to the plasmid DNA per transfection, serving as carrier to increase the transfection efficiency as previously described (Pradhan & Gadgil, 2012). 2 μl of jetPRIME was added, followed by 10-min incubation, before the mixture was added to the cells for 24 h before fixation.

Transient transfection for live cell imaging was performed using the Neon transfection system (Thermo Fisher Scientific). In short, $2–4 × 10^5$ cells were suspended in 100 μl buffer R of the Neon transfection kit, plasmid DNA (1.5–4 μg) was added, and following electroporation, parameters were used for pulse voltage, width and number: HeLa (1,005 V, 35 ms, 2#); U2OS (1,230 V, 10 ms, 4#).

Sequences of siRNA oligonucleotides were as follows: Mad2 (5′-ACCUUUACUCGAGUGCAGATTdTdT-3′; Nitta *et al*, 2004), control (targets luciferase; 5′-CGUACGCGGAAUACUUCGAdTdT-3′; Elbashir *et al*, 2001).

ZM447439 (Tocris Bioscience) was used at the indicated concentrations. RO-3306 (Tocris Bioscience) was used at 9 μM.

### Indirect immunofluorescence and microscopy

The following antibodies were utilised for indirect immunofluorescence: anti-CENP-C (R554; 1:500; Saitoh *et al*, 1992), anti-α-tubulin (DM1A; 1:500; Sigma-Aldrich), anti-HP1α (MAB3584; 1:200; Merck Millipore), anti-Aurora B (ab2254; 1:600; Abcam), anti-Aurora B (611082; 1:500; BD Transduction Laboratories; Figs EV2D and EV4A), anti-histone H3S10ph (06-570; 1:400; Merck Millipore), anti-histone H3T3ph (16B2; 1:500; H. Kimura) anti-cyclin A2 (6E6 ab16726; 1:100; Abcam), anti-cyclin B1 (GNS1 sc-245; 1:25; Santa Cruz Biotechnology), anti-Dsn1ph (1:1,000; a kind gift of Iain Cheeseman; Welburn *et al*, 2010), anti-Hec1 (9G3 ab3613; 1:500; Abcam).

Cells grown on coverslips were fixed in a pre-warmed 4% formaldehyde/PBS solution for 10 min and subsequently permeabilised in 0.5% Triton X-100/PBS for 10 min. Cells were blocked in 10% donkey serum/PBS for 1 h at room temperature before incubating with the individual primary antibodies at the stated concentrations. Secondary antibodies labelled with Alexa Fluor 488, 594 or 647 (Thermo Fisher Scientific) were diluted in PBS (1:400–1:1,000) and applied to the cells for 45 min. DNA was stained with Hoechst 33342, coverslips were mounted with ProLong Diamond Antifade (Thermo Fisher Scientific), and cured for 24 h before imaging. In experiments with pre-extraction, cells were incubated in pre-warmed 0.1% Triton X-100/PHEM buffer for 1 min before fixation. Pre-extraction buffer contained 1× PhosSTOP (Roche) in experiments with subsequent staining for Dsn1ph.

For cold-stable microtubule assays, 24 h after transfection cells were incubated in cooled Leibovitz's L-15 medium (Thermo Fisher Scientific) with 20 mM Hepes for 10 min on ice. Cells were fixed in

4% formaldehyde/PBS containing 0.2% Triton X-100 and stained as described above.

Optical sections of fixed cells were acquired every 0.2 μm using the CoolSNAP HQ CCD camera (Photometrics) on the wide-field microscope DeltaVision Spectris (Applied Precision) with a 100× NA 1.4 Plan Apochromat or 60× NA 1.4 PlanApo objective. SoftWoRx software (Applied Precision) was used for deconvolution. Images were imported into OMERO and adjusted with OMERO.figures for display (Allan *et al*, 2012). Shown images are maximum intensity projections.

### Live cell imaging

For live cell imaging experiments shown in Fig 2, cells were grown on glass-bottomed imaging chambers CG (Zell-Kontakt). Prior to imaging, the medium was changed to the $CO_2$-independent Leibovitz's L-15 medium (phenol red free), supplemented with 10% FBS, and the chamber lid was exchanged for a DIC lid (Zell-Kontakt). Live cell imaging was performed using the Eclipse Ti wide-field microscope (Nikon) with a Plan Apo λ 60× NA 1.4 objective and at 37°C in an environmental chamber. Optical sections were acquired every 2 μm using the ORCA-Flash 4.0 CMOS camera C11440-22CU (Hamamatsu), and 2 × 2 binning was applied to increase signal intensity. Expression levels of the CB-EY-HP1α constructs were determined according to the EYFP fluorescence values measured with Fiji (Schindelin *et al*, 2012). Maximum intensity values were determined in the movie frame in which cells underwent NEB. One region of interest (ROI) was applied to chromatin and the measured value subtracted by the average value of three ROIs applied to the cytoplasm. Fluorescence values from 300 to 1,000 were designated as low, > 1,000 to 3,000 medium, and > 3,000 to 6,000 high expression. Cells with fluorescence values below 300 and above 6,000 were excluded, because weak expression meant that mitotic defects like lagging chromosomes could not be determined reliably (as transfected cells could not be identified unambiguously) and very high expression caused the construct to localise to non-centromeric regions of the nucleus.

For live cell imaging using labelled Fab fragments, cells were grown in 35-mm glass bottom μ-dishes, high with DIC lid (ibidi). Bead loading of the Fab fragments (anti-histone H3S10ph: Fab313; anti-histone H3T3ph: Fab16B2; H. Kimura) was performed as described previously (Hayashi-Takanaka *et al*, 2011). Imaging was performed using the above described Eclipse Ti wide-field microscope with either the Plan Apo λ 60× NA 1.4 or Plan Apo 100× NA 1.40 objective. Movies were deconvolved using AutoQuant X3 (version X3.1.2) and projected by maximum intensity projection.

### Immunoblotting

Whole-cell lysates were prepared from HeLa cells transfected either with the indicated siRNAs or the indicated plasmids 24 h before harvesting. Membranes were incubated with primary antibodies recognising HP1α (15.19s2 05-689; 1:750; Merck Millipore; Fig 1D), GAPDH (ab9485; 1:2,500; Abcam; Fig 1D), α-tubulin (B512; 1:3,000; Sigma-Aldrich), Mad2 (A300-301A; 1:5,000; Bethyl), GFP (A-11122; 1:1,500; Thermo Fisher Scientific), HP1α (2616; 1:1,000; CST; Fig EV5A), HP1β (8876; 1:1,000; CST; Fig EV5A), HP1γ (MAB3450; 1:1,000; Millipore; Fig EV5A), GAPDH (016-25523; 1:10,000; Wako; Fig EV5A) and subsequently with IRDye 680rd or 800cw labelled secondary antibodies (LI-COR Biosciences) or HRP-linked secondary antibodies (GE Healthcare; Fig 1D). Fluorescence intensities were determined using the imaging systems Odyssey or Odyssey CLx (Fig EV5A; LI-COR Biosciences). HRP activity was determined after incubation with ECL substrate (Thermo Fisher Scientific) using the ChemiDoc MP imaging system (BioRad; Fig 1D).

### FRAP

Experiments for the various EYFP-fusion constructs were performed on a Leica SP5 confocal microscope equipped with an Argon laser using the 488 nm laser line and a 63×, 1.4 NA objective. HeLa cells grown on 25-mm round polylysine-coated coverslips were transfected 24 h before the experiments with the indicated plasmids using the above described jetPRIME protocol. Prior to the experiment, the medium was changed to FluoroBrite DMEM (without phenol red; Thermo Fisher Scientific) supplemented with 10% FBS. Temperature was maintained at 37°C in an environmental chamber (Life Imaging Services), and cells were gassed using 5% $CO_2$ in air. Five pre-bleach images were taken followed by bleaching a region of 1.6 μm diameter for 1 s at full laser intensity, selecting an ROI where the individual HP1 constructs clustered in interphase cells. Subsequent images were taken in three phases. The first rapid phase consisted of 20 frames every 0.65 s to observe the rapid initial recovery. The second phase consisted of 30 frames every 2 s to observe the slower stages of recovery. The third phase consisted of 45 frames every 5 s to ensure the complete steady state recovery was captured for all constructs.

Image data were processed using Image-Pro Premier (Media Cybernetics). The resulting intensity measurements were corrected for photobleaching and normalised according to Phair & Misteli (2001), whereby a ROI was applied to the bleach spot, background and non-bleached area of a nearby cell and compared to the five pre-bleach images. The $t_{1/2}$ values were calculated from the fluorescence values of 10 measurements after normalisation.

### Flow cytometry

HeLa cells were grown in 6-well plates and transfected with the indicated constructs or left untransfected. 24 h after transfection, cells were treated with either 3 μM ZM447439 or DMSO for 5 h, harvested and fixed in ice-cold 70% ethanol. Cells were washed in PBS with 0.05% Tween-20 and 1% BSA and stained with an anti-MPM2 antibody (ab14581, 1:400; Abcam) and subsequently an Alexa 647-labelled anti-mouse antibody (Thermo Fisher Scientific) to detect mitotic cells. Cells were incubated in PBS containing Hoechst 33342 (5 μg/ml) over night, and fluorescent cells were detected using an LSRII flow cytometer (BD Biosciences). Appropriate gates were set to evaluate transfected cells using the software FlowJo 8.7, and the percentage of MPM2-positive cells was determined within the whole population. Note that the varying mitotic index compared to Fig 1B is most likely due to the different methods of detection [microscopy (Fig 1B) vs. flow cytometry (Fig 3G)]. The flow cytometry analysis suggested a transfection efficiency of > 99%, in contrast to ~70% when analysed by eye using a fluorescence microscope, apparently because the flow cytometer can detect cells with very low expression

levels. Adjusting the gates of the flow cytometry experiment to 70% transfection efficiency of CB-EY-HP1 resulted in mitotic indices of 10.3% for CB-EY-HP1 with DMSO and 4.2% for CB-EY-HP1 with ZM447439.

### Automated image analysis of histone H3S10ph-positive $G_1$ cells

The Eclipse Ti wide-field microscope (Nikon) with a Plan Fluor 40× NA 1.3 objective was used to capture fixed cells. Optical sections were acquired every 0.7 μm using the ORCA-Flash 4.0 CMOS camera C11440-22CU (Hamamatsu). The numbers of histone H3S10ph-positive nuclei in $G_1$ cells were determined by an automated pipeline using the software CellProfiler (Kamentsky *et al*, 2011). The 3D data sets were projected (maximum intensity projection) and converted into TIFF files using standard CellProfiler modules. The analysis was performed using the following modules: IdentifyPrimaryObjects, MeasureObjectIntensity, ClassifyObjects and FilterObjects. The nuclei were identified using Hoechst 33342 staining. Transfected cells were identified based on the EYFP signal. Among the cyclin A2-negative nuclei (Alexa 647 fluorescence signal), the cells positive for histone H3S10ph were identified using the Alexa 594 signal. Detailed pipeline description is available upon request from Jan Ruppert.

### Automated image analysis of Dsn1ph signal

Fixed cells were captured using the DeltaVision wide-field microscope (see details above). Sum intensity projection after deconvolution was performed, and images were analysed using an adjusted standard CellProfiler pipeline. Following modules were used for segmentation based on the Hec1 signal (expanded by 1 pixel) and for measuring the mean Dsn1ph intensity: IdentifyPrimaryObjects, EnhanceOrSuppressFeatures, MaskImage, ExpandOrShrinkObjects. Some cells exhibited Dsn1ph intensity values with up to 600× the intensity of the median. To ensure that the results were not distorted by these extreme values, a general cut of value of 10 was applied in the same way in all experiments and repeats and cells excluded if they exhibited values above this cut-off. This resulted in following exclusions; CB-EY-HP1α: four cells; CB-EY-HP1αW174A: zero cell; untransfected cells: three cells.

### Statistical analysis

The Fisher's exact test was used for analysing data of the contingency table format. The Kolmogorov–Smirnov test was used for unpaired nonparametric data. The Benjamini–Hochberg procedure was used for all multiple testing corrections.

**Expanded View** for this article is available online.

### Acknowledgements

The authors thank Fusako Kawasaki and Iyo Toramoto (Kochi University) for their assistance. Flow cytometry experiments were performed in the Flow Cytometry Facility, Ashworth Laboratories, University of Edinburgh, and we thank Martin Waterfall for support. This work was funded by The Wellcome Trust, of which WCE is a Principal Research Fellow (Grant Number 107022). The Wellcome Trust Centre for Cell Biology is supported by core grant numbers 077707 and 092076 and Wellcome Trust Multi User Equipment Grant (WT104915MA). HK was supported by JSPS KAKENHI JP15K21730, JP25116005 and JP26291071. AAJ is supported by a Senior Research Fellowship from the Wellcome Trust (202811). SO was supported by Kato Memorial Bioscience Foundation and Takeda Science Foundation. JGR was supported by the Marie Curie Action PloidyNet, funded by the European Union Seventh Framework Programme (FP7/2007–2013) under Grant Agreement Number 607722.

### Author contributions

Conceptualisation, JGR, OM and WCE; Investigation JGR, KS, MP, HK, AAJ and SO; writing—original draft, JGR and WCE; writing—review and editing: JGR, OM, AAJ and WCE; visualisation, JGR and AAJ; supervision: WCE; funding acquisition, HK, AAJ, SO and WCE.

### Conflict of interest

The authors declare that they have no conflict of interest.

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
