## [Review Process File · The EMBO Journal]

HP1 α targets the chromosomal passenger complex for activation at heterochromatin before mitotic entry

Jan G. Ruppert, Kumiko Samejima, Melpomeni Platani, Oscar Molina, Hiroshi Kimura, A. Arockia Jeyaprakash Shinya Ohta and William C. Earnshaw

Review timeline:

Submission date:	26 June 2017
Editorial Decision:	27 July 2017
Additional correspondence - Authors	27 July 2017
Additional correspondence - Editor	27 July 2017
Appeal received:	3 August 2017
Editorial Decision:	11 August 2017
Revision received:	30 November 2017
Editorial Decision:	21 December 2017
Revision received:	24 January 2018
Accepted:	29 January 2018

Editor: Hartmut Vodermaier

Transaction Report:

1st Editorial Decision

27 July 2017

Thank you for submitting your manuscript on the effects of HP1 α tethering via CENP-B for our editorial consideration. It has now been reviewed and discussed by three expert referees, whose reports are copied below. I am afraid that say that in light of these comments, Bernd and I together came to conclude that the study is presently not a strong candidate for an EMBO Journal article, in the absence of (a) several important experimental controls whose outcome may profoundly affect key conclusions and interpretations of the study, and (b) follow-up utilization of the HP1 tethering system that may provide a more compelling and further-reaching advance over the current understanding on this topic. Since it is our policy to only invite revisions for those studies that receive significant interest and appear sufficiently close to being publishable already after the first round of review, the above-mentioned major technical and conceptual reservations unfortunately preclude us from offering concrete further proceedings at The EMBO Journal in this case. Thank you in any case for having had the opportunity to consider this work. I am sorry that we cannot be more positive on this occasion, but nevertheless hope that you will find our referees comments and suggestions helpful.

 REFEREE REPORTS

Referee #1:

It has been long known that HP1 dissociates from the heterochromatin histone mark H3K9me3 upon Aurora B-dependent phosphorylation of H3S10 during mitosis, but its functional significance is largely speculative. By tethering HP1 to the centromere via CENP-B fusion, Earnshaw and his

colleagues here now demonstrate that HP1 dissociation is required to allow the CPC to dynamically change its localization from the inner centromere to the spindle midzone during the course of mitosis. If HP1 is tethered to the centromere, INCENP, which is known to directly bind to HP1, is targeted to the centromere, leading to the stable localization of the CPC to the centromere throughout the mitotic progression and even in telophase and G1 phase cells, where H3S10 is remain phosphorylated at the centromere. Forced enrichment of HP1 by this method abortively activates the spindle assembly checkpoint and arrest cells in metaphase, consistent with a published observation on cells expressing INCENP tethered to CENP-B.

Overall, the authors performed the majority of experiments in a carefully controlled manner, and results support the authors' major conclusion. Although I have a modest reservation of authors' novelty claim, I support publication of this manuscript after minor revisions listed below.

1. One of the major importance of this manuscript is to provide evidence that HP1 dissociation from mitotic chromosomes is required for dynamic localization change of the CPC. Although the authors indeed demonstrated this point by elegant experiments, I am not comfortable of the claim described in the last sentence of Discussion, "The suggestion that HP1 is an auxiliary member of the CPC during early mitosis for the first time reveals the importance of H3S10 phosphorylation and the methyl/phos switch on HP1 binding for ensuring the normal dynamics of CPC movements during mitosis."

Nozawa et al (2010) actually demonstrated that HP1 recruits INCENP to heterochromatin in interphase, and upon inhibition of Aurora B, INCENP is mislocalized to chromosome arms in a manner dependent on HP1-INCENP interaction (Fig. S2 and S5). These observations have already hinted the importance of HP1 dissociation in CPC localization control. I feel that it is appropriate to discuss these data, and soften the priority claim.

2. Figure 1. I assume that fixed cells were analyzed in this experiment, and cells were counted regardless of transfection levels. If so, it would be important to comment on transfection efficiency, although I understand that better live imaging analysis supported the authors' conclusion.

3. Figure 6. I am curious to know if H3S10ph is maintained throughout the interphase beyond G1. Although the authors note that H3S10ph is normally observed in G2, robust phosphorylation can be seen just prior to prophase, and level of H3S10ph is almost undetectable during S and early G2 in cancer cell lines. See Hayashi-Takanaka et al 2009 (PMID: 19995936).

4. Page 19. "Thus, when the CPC is held at G1 centromeres by HP1, the Aurora B kinase activity is sufficient to counteract any conflicting phosphatase activity, possibly through continuous intermolecular self-activation at these clusters (Sessa et al, 2005)."
Also cite Kelly et al 2007 (PMID 17199039), which experimentally demonstrates the clustering-induced activation of Aurora B.

Referee #2:

In the manuscript "Heterochromatin protein 1 α dynamics regulate chromosomal passenger complex activity at centromeres", authors Ruppert, Earnshaw and colleagues engineer a variety of centromere-targeted HP1 α constructs and express them in HeLa and U2OS cells to determine the role of HP1 α release from chromatin in CPC regulation. The experiments demonstrate that constitutive targeting of HP1 α to centromeres results in a severe cell cycle delay that is dependent on the chromoshadow and hydrophobic pocket domains of HP1 α as well as the SAC. Although the authors create a few molecular tools with some potential, I feel that these tools were not used to sufficiently advance our knowledge of the role of HP1 α in CPC localization and function. Unfortunately, I do not believe that the conclusions drawn by the authors are of significant enough impact to be published in The EMBO Journal. In addition, I have many concerns about the nature of the experiments and the lack of controls as outlined below. I do, however, propose a suggestion for how the authors could potentially use some of the tools that they have created to address a more pressing question that would likely be of high interest in the final section of this review.

Major concerns:

1. It is unclear where precisely CENP-B conjugation targets fusion proteins. The authors claim that it tethers HP1alpha to "its normal location in the inner centromere". However, CENP-B is generally considered to be a centromeric protein, not an inner-centromeric protein. Although the authors are correct in citing that CENP-B has been observed "throughout the centromeric region and beneath the kinetochore" in some cases, this does not necessarily qualify as "its normal localization". Evaluating potential differences between HP1alpha and CB-HP1alpha are made especially difficult by the lack of a direct comparison of localization for the CB-EY-HP1alpha and EY-HP1alpha constructs. This would preferably be on cells with lower amounts of over-expression. Additionally, line scans across the two kinetochores would be helpful to get a better idea of precise localization.

The complications with discerning between centromeric and inner-centromeric localization are highlighted by the recent publication Hengeveld et al. 2017 (DOI: 10.1038/ncomms15542). Initial work published by the same group interpreted the chromosome missegregation phenotype of CB-INCENP-expressing cells as resulting from increased/constitutive activity near the kinetochore. However, their recent results suggest that the CB-INCENP construct actually moves the CPC from the inner-centromere to the centromere, resulting in a cohesion defect. To demonstrate that a similar unintended consequence is not occurring for the CB-EY-HP1alpha construct, it would be good to demonstrate that the phenotype that they observe is not from cohesin disruption, perhaps by looking at chromosome spreads.

2. To demonstrate the specificity of the metaphase delay phenotype for Aurora B over-activation, the authors look at the mitotic phases of CB-EY-HP1alpha-expressing cells in the presence of the Aurora B inhibitor ZM447439 (Figure S4D). This is an absolutely essential control to demonstrate the specificity of the phenotype from perturbation of HP1 localization. Unfortunately, the shift in mitotic phase upon addition of the inhibitor appears to be dominated by the anaphase defects associated with Aurora B inhibition. 40-50% of cells appear to be in anaphase in the ZM treatment compared to less than 10% for controls (e.g. Fig. 1C). It is therefore impossible to determine if the decrease in metaphase cells is due to recovery of the spindle checkpoint arrest or due to the increase in anaphase and telophase cells. Minimally, measuring of the timing of metaphase following drug treatment would be necessary to distinguish between these two possibilities. Addition of a ZM-treated control without CB-HP1alpha would also be helpful.

3. Measuring the degree of expression for the fusion constructs relative to wild-type is essential to interpreting these results. Artificial mislocalization of a protein is already difficult to extrapolate back to its ordinary function. Artificial mislocalization combined with overexpression would make the results even more difficult to interpret. Although the authors do a good job of showing that the different mutant constructs are expressed at similar levels, it is unclear how this compares to the endogenous HP1. A western with an HP1alpha antibody would help with this.

4. As with any artificially-targeting construct, it is imperative to establish if the observed phenotypes associated with the fusion are due to targeting the protein to the desired location, and not removing them from another location. This is especially important for proteins that are known to oligomerize, such as HP1, where the chimeric protein could strip the endogenous from an important location within the cell. The W174A mutant is a good control for this, as it eliminates the phenotype while presumably leaving the dimerization capabilities of HP1 intact. However, given the potentially high degree of overexpression of the fusion construct (see above), it still seems quite plausible that the endogenous HP1 would be affected through the PxVxL-binding motif. It would therefore be good if the authors could either demonstrate or cite clear differences between the CB-EY-HP1alpha phenotype and an HP1 depletion phenotype.

5. The authors conclude in their abstract "Our results reveal that H3S10 phosphorylation is essential to release HP1alpha and allow CPC mobility during mitosis." It is unclear how this statement comes from the results in the manuscript, as there are no experiments that disrupt H3S10 phosphorylation and look at HP1alpha release.

6. One of the authors' other main conclusions is that "it may be useful to consider HP1alpha as an auxiliary member of the CPC during early mitosis". I fail to see how this is useful. Distinguishing between a complex "member" or "auxiliary subunit" vs. a strong interactor as previously demonstrated

(e.g. Abe et al. 2016) seems largely semantic.

Minor:

The authors state that "the centromeric localisation of these mutants was indistinguishable from that of CB-EY-HP1 α containing wild type HP1 α (Fig. S2B-E)." This statement requires quantification of intensity and localization, as some of the images look quite different from each other (the I165E mutant appears to have decreased localization and more background). Similarly, the claimed differences in Aurora B localization in Figure S4A-C are unclear and require quantification.

Suggestion:

In the discussion, the authors write "Interestingly, no metaphase delay is seen when HP1 α is tethered by a CENP-BDBD mutant that has impaired DNA binding and causes a roughly 3-fold increase in HP1 α dynamics at centromeres." This could indeed be interesting in light of recent work that demonstrates a decrease in HP1 localization associated with CIN in cancer cell lines, and suggests that decreased HP1 at centromeric regions could be a cause of CIN. If the authors use the CBmut-EY-HP1 α tool to further test this hypothesis and determine if extra HP1 α activity rescues CIN, it would greatly increase the impact of the manuscript.

Referee #3:

The study here investigates a role for HP1 α in the regulation of Chromosomal Passenger Complex (CPC) dynamics at the centromere during mitosis. The authors constitutively tether HP1 α to centromeres by fusing it directly to the DNA binding domain of CENPB. This results in (1) a spindle assembly checkpoint (SAC)-mediated mitotic arrest, and (2) inhibition of CPC relocation from centromeres to the midbody at anaphase. This study nicely demonstrates that the status of HP1 α binding to chromatin is able to dictate the recruitment and eviction of the CPC from centromeres. My only concern is the authors' interpretation of the results regarding the role of the CPC in SAC activity. Upon constitutively tethering HP1 α to centromeres via CENPB, they observe a robust mitotic arrest that is Mad2-dependent. They go on to conclude that the mitotic arrest is due to perturbation of CPC dynamics which prevent the CPC from participating in its normal role in SAC signaling. A more likely explanation for the mitotic arrest is that when Aurora B kinase is constitutively localized to the centromere region, it continues to phosphorylate mitotic substrates which results in unstable kinetochore-MTs and an expected Mad2-dependent mitotic delay/arrest. The authors should check if this is the case by assessing kinetochore-MT attachment stability (cold-stable MT assay, level of k-k stretch, etc) and measuring the phosphorylation of known Aurora kinase B substrates.

Additional comment:

-The authors state that CENP-B-HP1 α tethering results in CPC recruitment to the chromatin independently of pH3 and pH2A. They should formally test this by inhibiting Bub1 and Haspin and assessing if CENPB-HP1 α is still able to recruit the CPC.

Additional Correspondence – Authors' response

27 July 2017

Thank you for overseeing the review process for our MS. Obviously I am disappointed with the outcome, but I am also slightly confused by the reviews and your response to them.

I understand that we are not privy to the confidential comments and discussions of the referees, but I cannot help but gain the impression that Referees 1 and 3 have relatively minor concerns with the MS, while referee 2 has stated a number of more serious concerns. Your response seems to show that you place more weight on the comments of referee 2 and assume that we are unable to answer most or all of the points raised. However, we feel that we could indeed answer the majority of those points (some of which are essentially semantic and could be addressed by re-wording). It therefore seems that there may be information that we are not privy to that has swung the balance against allowing us to revise our MS. If this is the case, then that is fair enough.

I guess what I would like a more clear statement on is whether you would allow us to revise the MS in response to referee 2 or whether you feel that even if we could answer those comments, you would not be interested in seeing a revision. Sorry, I am not entirely certain about what "preclude us from offering concrete further proceedings at The EMBO Journal" means. Does that mean that we could try again with no guarantee of ultimate acceptance?

Thank you again for considering the MS and I am sorry to take more of your time with this, but would appreciate a brief response clarifying the situation.

Additional correspondence – Editor's response

27 July 2017

Thank you for your response, and apologies if I may not have been very clear in my decision letter.

The main reason for which we decided not to invite revision is the moderate overall advance/novelty, as stated by both referees 1 and 2 (even if referee 1 indicates they would still support publication, as you rightly notice). Referee 2 makes a suggestion that could in their view increase the overall impact of the work, but since this is a further-reaching proposal exceeding the scope of the manuscript as currently presented, we did not see it justified to request such extension work. For the sake of fairness to authors, we only invite revisions if we feel that we can more or less commit to a paper and its eventual publication, not in cases where a study would need further extension and the outcome is not clear.

Beyond that, the experimental issues raised by the referees (expression levels, nature of CENP-B-mediated targeting, exact basis of the observed phenotype of HP1 tethering...) would seem like essential prerequisites for any further consideration. Again, there is no secret communication from the referees here (we abolished confidential comments to the editor some 6 years ago), but in their post-review discussions referees 2 and 3 simply reiterated that to support the key conclusion that release of HP1 from mitotic chromosomes by H3S10p is important,

"...we need some evidence that the levels, stability, and location of HP1 targeted to the chromatin by CENP-B tethering resemble what would be created by inhibition of H3S10 phosphorylation. Stability could potentially be accomplished by performing their FRAP assay from Figure 5 on EY-HP1alpha with Aurora B inhibition. Expression levels and localization were already addressed as concerns in my review.

Second, to establish that the HP1 release is important, there needs to be some phenotype associated with it. Currently, the only connection between the inability to release HP1 and a phenotype is the metaphase arrest. Metaphase arrest is also the main assay for the first 4 figures. I therefore maintain that any issues with this assay are crucial to address." [ref 2]

"I agree that the demonstrating kinetochore-MT attachment instability is not necessarily novel/exciting, however this is the likely cause of the mitotic arrest. The authors seem to be concluding that constitutive tethering of HP1 leads to a defect in SAC signaling per se, rather than a defect in K-MT stability, which leads to an expected mitotic arrest through a properly-functioning SAC. I think it is important that they clarify this point." [ref 3]

So, to answer your question whether we would be open to looking at a new version of this study once more, I would say in principle Yes, but I think it would be important to discuss beforehand in which way you might be able to respond to the referees' concerns and our reservations. In this case, it would be helpful if you could prepare a tentative point-by-point response and revision proposal as a basis for such consideration, which we might also discuss with some of the referees if necessary.

I hope this helps to clarify, and look forward to hearing what you think about this.

Additional correspondence – Authors' response

3 August 2017

WELLCOME TRUST CENTRE *for* CELL BIOLOGY
INSTITUTE OF CELL BIOLOGY, UNIVERSITY OF EDINBURGH

MICHAEL SWANN BUILDING, KING'S BUILDINGS, MAX BORN CRESCENT, EDINBURGH EH9 3BF, SCOTLAND

William C. Earnshaw, Ph.D., FRS, FRSE, FMedSci
Professor and Principal Research Fellow of the Wellcome Trust
phone +44 - (0)131 - 650-7101
FAX +44 - (0)131 - 650-7100
E-mail: bill.earnshaw@ed.ac.uk

**Chromosome
Structure
Lab**

August 3, 2017

Hartmut Vodermaier, PhD
Senior Editor / The EMBO Journal
h.vodermaier@embojournal.org
Edinburgh EH9 3BF

Re: EMBOJ-2017-97677

Dear Hartmut:

Thank you very much for agreeing to reconsider your decision on our MS, EMBOJ-2017-97677. Jan and I have gone carefully over the many insightful comments of the referees. We were extremely pleased that referees 1 and 3 were basically supportive of publication in the EMBO Journal, and hope that we are able to adequately address the many concerns of referee 2.

We believe that it should be straightforward to address all issues of controls, and describe in our response below just how we would do this. Certain of the concerns can be explained by rewording or expanding explanations in the text, and we apologise if we were unclear in our original presentation of the work. Overall, we believe that there were no issues that we could not address within a reasonable time.

The only outstanding issue that we must address is the novelty of the MS. We believe that this recognition that interaction with HP1 is an important determinant of CPC localisation is important and novel in the sense that basically the entire field believes that CPC localisation is driven solely by H3T3 phosphorylation and the H2AT120ph/SGO1 axis. Our work shows that in addition to those mechanisms, HP1 also has an important role to play. We also believe that our observation that the CPC clustered at centromeres is catalytically active throughout the cell cycle is entirely novel and provocative as it raises important issues about the regulation of CPC activity. When we have completed the live-cell Fab tracking experiment suggested by Referee 1, this will add another figure and further bolster the novelty of the MS.

We attach below a response to the comments of the referees which we would be grateful if you would consider. We will look forward to hearing from you when you have had an opportunity to go over this response and make a decision.

Best regards,

Referee #1:

It has been long known that HP1 dissociates from the heterochromatin histone mark H3K9me3 upon Aurora B-dependent phosphorylation of H3S10 during mitosis, but its functional significance is largely speculative. By tethering HP1 to the centromere via CENP-B fusion, Earnshaw and his colleagues here now demonstrate that HP1 dissociation is required to allow the CPC to dynamically change its localization from the inner centromere to the spindle midzone during the course of mitosis. If HP1 is tethered to the centromere, INCENP, which is known to directly bind to HP1, is targeted to the centromere, leading to the stable localization of the CPC to the centromere throughout the mitotic progression and even in telophase and G1 phase cells, where H3S10 is remain phosphorylated at the centromere. Forced enrichment of HP1 by this method abortively activates the spindle assembly checkpoint and arrest cells in metaphase, consistent with a published observation on cells expressing INCENP tethered to CENP-B.

Overall, the authors performed the majority of experiments in a carefully controlled manner, and results support the authors' major conclusion. Although I have a modest reservation of authors' novelty claim, I support publication of this manuscript after minor revisions listed below.

We thank the referee for these thoughtful and positive comments and for supporting the publication of our manuscript in the EMBO Journal

1. One of the major importance of this manuscript is to provide evidence that HP1 dissociation from mitotic chromosomes is required for dynamic localization change of the CPC. Although the authors indeed demonstrated this point by elegant experiments, I am not comfortable of the claim described in the last sentence of Discussion, "The suggestion that HP1 is an auxiliary member of the CPC during early mitosis for the first time reveals the importance of H3S10 phosphorylation and the methyl/phos switch on HP1 binding for ensuring the normal dynamics of CPC movements during mitosis."

Nozawa et al (2010) actually demonstrated that HP1 recruits INCENP to heterochromatin in interphase, and upon inhibition of Aurora B, INCENP is mislocalized to chromosome arms in a manner dependent on HP1-INCENP interaction (Fig. S2 and S5). These observations have already hinted the importance of HP1 dissociation in CPC localization control. I feel that it is appropriate to discuss these data, and soften the priority claim.

We thank the referee for highlighting the results from the supplementary data of Nozawa et al and we are happy to include these observations in our discussion, as they are supportive for our findings on HP1 regulation of CPC localisation.

2. Figure 1. I assume that fixed cells were analyzed in this experiment, and cells were counted regardless of transfection levels. If so, it would be important to comment on transfection efficiency, although I understand that better live imaging analysis supported the authors' conclusion.

The referee noted correctly that in our initial experiments, in which the mitotic delay was identified, cells were analysed independent of the expression level of the transiently transfected proteins. To perform a more detailed analysis of the mitotic delay, we subsequently used live cell imaging in Figure 3 and there, we carefully took the expression levels into account.

To clarify the experimental setup of the initial experiments, we are happy to report the transfection efficiency.

3. Figure 6. I am curious to know if H3S10ph is maintained throughout the interphase beyond G1. Although the authors note that H3S10ph is normally observed in G2, robust phosphorylation can be seen just prior to prophase, and level of H3S10ph is almost undetectable during S and early G2 in cancer cell lines. See Hayashi-Takanaka et al 2009 (PMID: 19995936).

We thank the referee for pointing out this question and for their further interest in the robustness of the H3S10ph mark in interphase. Analysis of fixed cells stained for Cyclin A2 or Cyclin B suggests that the H3S10ph mark is indeed robust through interphase.

An elegant way to answer this question and directly follow the robustness of the H3S10ph mark throughout interphase could be the use of the Fab antibody fragments from the above-

mentioned Hayashi-Takanaka et al publication. Hiroshi Kimura is a collaborator of ours who already shared the required reagents with us and taught Jan Ruppert how to load them into growing cells during a visit to his lab in Tokyo. Thus, we are confident that we could do this novel experiment.

4. Page 19. "Thus, when the CPC is held at G1 centromeres by HP1, the Aurora B kinase activity is sufficient to counteract any conflicting phosphatase activity, possibly through continuous intermolecular self-activation at these clusters (Sessa et al, 2005)."

Also cite Kelly et al 2007 (PMID 17199039), which experimentally demonstrates the clustering-induced activation of Aurora B.

We thank the referee for highlighting the work of Kelly et al 2007 as an helpful addition to our discussion and we are happy to include this in the manuscript.

Referee #2:

In the manuscript "Heterochromatin protein 1 α dynamics regulate chromosomal passenger complex activity at centromeres", authors Ruppert, Earnshaw and colleagues engineer a variety of centromere-targeted HP1 α constructs and express them in HeLa and U2OS cells to determine the role of HP1 α release from chromatin in CPC regulation. The experiments demonstrate that constitutive targeting of HP1 α to centromeres results in a severe cell cycle delay that is dependent on the chromoshadow and hydrophobic pocket domains of HP1 α as well as the SAC. Although the authors create a few molecular tools with some potential, I feel that these tools were not used to sufficiently advance our knowledge of the role of HP1 α in CPC localization and function. Unfortunately, I do not believe that the conclusions drawn by the authors are of significant enough impact to be published in The EMBO Journal. In addition, I have many concerns about the nature of the experiments and the lack of controls as outlined below. I do, however, propose a suggestion for how the authors could potentially use some of the tools that they have created to address a more pressing question that would likely be of high interest in the final section of this review.

We thank the referee for the comments on our manuscript and hope to address all concerns and ambiguities with our comments

Major concerns:

1. It is unclear where precisely CENP-B conjugation targets fusion proteins. The authors claim that it tethers HP1 α to "its normal location in the inner centromere". However, CENP-B is generally considered to be a centromeric protein, not an inner-centromeric protein. Although the authors are correct in citing that CENP-B has been observed "throughout the centromeric region and beneath the kinetochore" in some cases, this does not necessarily qualify as "its normal localization".

We appreciate the referee's concern about the targeting via the CENP-B DNA binding domain. We believe that the concerns about the localisation of CENP-B are arising from inaccuracy in literature not distinguishing between the centromeric and inner centromeric positioning of *ENDOGENOUS* CENP-B, which was reported (based on immunogold pre-embedding EM) as extending from the inner kinetochore down through the central domain of the centromere (PMID: 2335558 – see also the double label fluorescence microscopy of endogenous CENP-B and CENP-C in PMID: 9382805). In our view, these careful studies of the localisation of the endogenous protein are more reliable than numerous light microscopy studies of exogenously expressed CENP-B and CENP-B fragments. Our view is that CENP-B is located in the inner kinetochore, but also in the heterochromatin of the inner centromere.

Furthermore, the precise localisation of CENP-B depends on the DNA methylation status of the CENP-B box, methylation of which can prevent CENP-B binding. Therefore, different localisation of CENP-B is possible among different cell lines or even different clones of the same cell line. We believe that the safest response to this concern is simply to remove the statement about "its normal localization", although we do believe that we are targeting HP1 to the correct domain in our clone of HeLa cells.

Evaluating potential differences between HP1alpha and CB-HP1alpha are made especially difficult by the lack of a direct comparison of localization for the CB-EY-HP1alpha and EY-HP1alpha constructs. This would preferably be on cells with lower amounts of over-expression. Additionally, line scans across the two kinetochores would be helpful to get a better idea of precise localization.

The complications with discerning between centromeric and inner-centromeric localization are highlighted by the recent publication Hengeveld et al. 2017 (DOI: 10.1038/ncomms15542). Initial work published by the same group interpreted the chromosome missegregation phenotype of CB-INCENP-expressing cells as resulting from increased/constitutive activity near the kinetochore. However, their recent results suggest that the CB-INCENP construct actually moves the CPC from the inner-centromere to the centromere, resulting in a cohesion defect. To demonstrate that a similar unintended consequence is not occurring for the CB-EY-HP1alpha construct, it would be good to demonstrate that the phenotype that they observe is not from cohesin disruption, perhaps by looking at chromosome spreads.

Hengeveld et al. 2017 and the preceding work by the Lens group (Liu et al. 2009) use the DNA binding domain of CENP-B fused to INCENP. In that very complex and elegant paper, when describing the localisation of their fusion proteins, the authors state “Although some Aurora B remained associated with centromeric heterochromatin in CB-INCENP expressing cells, we will refer to this CB-INCENP-shifted pool of Aurora B as ‘kinetochore proximal’”. Thus, the localization of Aurora B in their experiments is complex, and observed phenotypes could be due to altered dynamics, and not localisation. We raise this point because, in another important caveat, the authors apparently do not consider that Survivin-INCENPdeltaCen will bind dynamically to chromatin based on recognition of histone marks, whereas CENP-B-DBD-INCENPdeltaCen will bind more stably to DNA wherever CENP-B boxes are not methylated.

The CENP-B DNA-binding domain is commonly used to target proteins to centromeres, but as used by Hengeveld et al. 2017, it does not contain the CENP-B dimerization domain and also INCENP does not facilitate dimerization. Importantly, our experiments reveal that CENP-B dimerization is required for stable binding of this protein at centromeres. For CB-EY, which consists of only the DNA-binding domain and EYFP without any fused protein, we measured a half time of recovery in FRAP experiments that was ~ 7 times faster than the binding dynamics of the chimeric CB-EY-HP1 α construct and was very similar to the kinetics seen with CB-EY-HP11165E, which does not dimerise. Thus in our experiments, dimerization was provided by the HP1 moiety. We believe it is possible that dimerization is required for CENP-B to localise normally in the inner centromere, as the endogenous protein does in HeLa cells (PMID: 2335558).

Additionally, Hengeveld et al. 2017 saw the above-mentioned cohesion defect only when the endogenous INCENP was depleted and CB-INCENP was expressed (see their Figure 2B). However, in our experiments we do not deplete any endogenous proteins when assessing the effect of tethered HP1.

We propose to make following changes to address the referee’s concerns:

- Re-phrasing of the HP1 tethering location to: “targeting HP1 to the centromere”
- Show the localisation of EY-HP1 compared to CB-EY-HP1 and include line scans as suggested by the referee. We will try to do this on mitotic chromosomes, but note that this localisation may have to be shown in interphase, since due to the methyl-phos switch on H3S10, most HP1 is displaced from mitotic chromosomes.

2. To demonstrate the specificity of the metaphase delay phenotype for Aurora B over-activation, the authors look at the mitotic phases of CB-EY-HP1alpha-expressing cells in the presence of the Aurora B inhibitor ZM447439 (Figure S4D). This is an absolutely essential control to demonstrate the specificity of the phenotype from perturbation of HP1 localization. Unfortunately, the shift in mitotic phase upon addition of the inhibitor appears to be dominated by the anaphase defects associated with Aurora B inhibition. 40-50% of cells appear to

be in anaphase in the ZM treatment compared to less than 10% for controls (e.g. Fig. 1C). It is therefore impossible to determine if the decrease in metaphase cells is due to recovery of the spindle checkpoint arrest or due to the increase in anaphase and telophase cells. Minimally, measuring of the timing of metaphase following drug treatment would be necessary to distinguish between these two possibilities. Addition of a ZM-treated control without CB-HP1alpha would also be helpful.

We apologise if the experiment was not described properly and caused confusion. In this experiment we treated the cells with ZM for only 30 minutes before fixation. Therefore, the high amount of cells in anaphase and telophase cells results from the sudden release of cells from the metaphase delay that occurred during the 24 h of transient transfection and not from a gradual accumulation of cells blocked in mitotic exit. Many studies of the CPC show that when the complex is inhibited the phenotype is not an accumulation of cells in ana/telophase, but instead an accumulation of binucleated cells.

We thank the referee for pointing out an additional control and will include untransfected cells treated with ZM in the experiment.

3. Measuring the degree of expression for the fusion constructs relative to wild-type is essential to interpreting these results. Artificial mislocalization of a protein is already difficult to extrapolate back to its ordinary function. Artificial mislocalization combined with overexpression would make the results even more difficult to interpret. Although the authors do a good job of showing that the different mutant constructs are expressed at similar levels, it is unclear how this compares to the endogenous HP1. A western with an HP1alpha antibody would help with this.

We note that to address this concern we very specifically segregated our critical results of live imaging according to whether cells were expressing low, medium or high levels of the chimeric proteins. We believe that it is more accurate to judge the level of expression at the single cell level by fluorescence microscopy than to take the average for the culture as a whole. However to address this concern, we will include a Western blot staining with an anti-HP1alpha antibody.

4. As with any artificially-targeting construct, it is imperative to establish if the observed phenotypes associated with the fusion are due to targeting the protein to the desired location, and not removing them from another location. This is especially important for proteins that are known to oligomerize, such as HP1, where the chimeric protein could strip the endogenous from an important location within the cell. The W174A mutant is a good control for this, as it eliminates the phenotype while presumably leaving the dimerization capabilities of HP1 intact. However, given the potentially high degree of overexpression of the fusion construct (see above), it still seems quite plausible that the endogenous HP1 would be affected through the PxVxL-binding motif. It would therefore be good if the authors could either demonstrate or cite clear differences between the CB-EY-HP1alpha phenotype and an HP1 depletion phenotype.

We thank the referee for noting the utility of using the W174A mutant as a control, but have difficulties understanding how “endogenous HP1 would be affected through the PxVxL-binding motif”. Neither the CENP-B-DBD nor HP1 α contains a PxVxL motif and studies like that by Nozawa et al. show that HP1 dimerisation is not affected by the W174A mutation.

Additionally, publications that deplete HP1 do not describe phenotypes like we show in our work. Looking through the literature we find the following examples of phenotypes following HP1 depletion:

1. less tightly paired sister chromatids and delocalization of Aurora B
2. overcondensed chromosomes in which the distance between the sister chromatids, both at arms and at centromeres, was clearly increased with respect to control chromosomes and the centromeric Aurora B signal was lost
3. multipolar spindles
4. pseudoanaphase with scattered chromosomes along the elongated spindle

5. a bipolar spindle in which a number of chromosomes (from one to five) had not congressed yet to the metaphase plate

In chromosome spreads, we note, if anything, a tightening of the association between sister chromatids in cells expressing CB-HP1. The chromosome arms are not over-condensed. We also did not notice difficulties with chromosome congression, as shown in the example of the live cell imaging experiment – where a well organised metaphase plate is formed within 12 minutes after NEB (Figure 3C). We therefore believe that the W174A mutant is a suitable control, as recognised by the referee.

5. The authors conclude in their abstract "Our results reveal that H3S10 phosphorylation is essential to release HP1alpha and allow CPC mobility during mitosis." It is unclear how this statement comes from the results in the manuscript, as there are no experiments that disrupt H3S10 phosphorylation and look at HP1alpha release. We apologise for any unclarity and will rephrase this section also in light of the supplementary data of Nozawa et al as suggested by referee no.1

6. One of the authors' other main conclusions is that "it may be useful to consider HP1alpha as an auxiliary member of the CPC during early mitosis". I fail to see how this is useful. Distinguishing between a complex "member" or "auxiliary subunit" vs. a strong interactor as previous demonstrated (e.g. Abe et al. 2016) seems largely semantic.

We thank the referee for pointing out the work from Abe et al who state in their abstract "we show that heterochromatin protein 1 (HP1) is an essential CPC component required for full Aurora B activity." This further supports our finding for the importance of HP1 for the CPC and we will include this in our manuscript in more detail, citing Abe et al for suggesting this idea first.

This point is important because e.g. in structural studies where various laboratories have had difficulties in expressing the soluble CPC complex, it is possible that the addition of HP1alpha may allow the production of suitable samples for future study.

Minor:

The authors state that "the centromeric localisation of these mutants was indistinguishable from that of CB-EY-HP1 α containing wild type HP1 α (Fig. S2B-E)." This statement requires quantification of intensity and localization, as some of the images look quite different from each other (the I165E mutant appears to have decreased localization and more background).

We thank the referee for highlighting the discrepancy between supplementary figure S2 and the corresponding text. The CB-EY-HP1-I165E mutant shows much faster dynamics compared to the other CB-EY-HP1 constructs as we describe in detail in the discussion. This is most likely the reason for the increased background signal. This is one reason why we avoided using this mutant as a control construct, although it does localise to centromeres. We will improve the representation and rephrase the manuscript to increase clarity.

Similarly, the claimed differences in Aurora B localization in Figure S4A-C are unclear and require quantification. To increase the clarity for the difference in Aurora B localisation we will perform a quantification of the Aurora B signal on chromosome arms.

Suggestion:

In the discussion, the authors write "Interestingly, no metaphase delay is seen when HP1 α is tethered by a CENP-BDBD mutant that has impaired DNA binding and causes a roughly 3-fold increase in HP1 α dynamics at centromeres." This could indeed be interesting in light of recent work that demonstrates a decrease in HP1

localization associated with CIN in cancer cell lines, and suggests that decreased HP1 at centromeric regions could be a cause of CIN. If the authors use the CBmut-EY-HP1 α tool to further test this hypothesis and determine if extra HP1 α activity rescues CIN, it would greatly increase the impact of the manuscript.

We thank the referee for this interesting suggestion. We already assessed the effect of tethered HP1 on chromosome segregation defects and found an increase upon CB-EY-HP1 expression. However, the effect of weakly tethered HP1 on chromosome segregation could be an interesting approach. To test this we propose to express CBmut-EY-HP1 in U2OS cells, which show a base line value of lagging chromosomes of 27 % and are therefore well suited for this experiment. Staining with SiR-DNA dye will allow us to visualize missegregation events in live cell imaging and allow the exact identification and quantification of an improved chromosome mis-segregation rate.

We recognise that if CBmut-HP1 actually causes a decrease in CIN in U2OS cells, then this would make a very interesting addition to the MS and could be an opening for a larger study in the future.

Referee #3:

The study here investigates a role for HP1 α in the regulation of Chromosomal Passenger Complex (CPC) dynamics at the centromere during mitosis. The authors constitutively tether HP1 α to centromeres by fusing it directly to the DNA binding domain of CENPB. This results in (1) a spindle assembly checkpoint (SAC)-mediated mitotic arrest, and (2) inhibition of CPC relocation from centromeres to the midbody at anaphase. This study nicely demonstrates that the status of HP1 α binding to chromatin is able to dictate the recruitment and eviction of the CPC from centromeres. My only concern is the authors' interpretation of the results regarding the role of the CPC in SAC activity. Upon constitutively tethering HP1 α to centromeres via CENPB, they observe a robust mitotic arrest that is Mad2-dependent. They go on to conclude that the mitotic arrest is due to perturbation of CPC dynamics which prevent the CPC from participating in its normal role in SAC signaling. A more likely explanation for the mitotic arrest is that when Aurora B kinase is constitutively localized to the centromere region, it continues to phosphorylate mitotic substrates which results in unstable kinetochore-MTs and an expected Mad2-dependent mitotic delay/arrest. The authors should check if this is the case by assessing kinetochore-MT attachment stability (cold-stable MT assay, level of k-k stretch, etc) and measuring the phosphorylation of known Aurora kinase B substrates.

We thank the referee for the supportive review and for the suggestions how to improve our manuscript in terms of the mitotic delay mechanism. Measuring phosphorylation of the Aurora B substrate Dsn1 and the cold-stable MT assay should help to gain further insight into the mechanistic cause of the mitotic delay.

Additional comment:

-The authors state that CENP-B-HP1 α tethering results in CPC recruitment to the chromatin independently of pH3 and pH2A. They should formally test this by inhibiting Bub1 and Haspin and assessing if CENPB-HP1 α is still able to recruit the CPC.

We appreciate that the referee is pointing out the possibility of CPC recruitment via H3T3 and H2AT120. These marks are playing crucial roles in CPC recruitment to the centromere but have not been described to play a role in CPC localisation upon anaphase onset. However, with the tethered HP1 we observe a substantial amount of Aurora B remaining at centromeres even in telophase and later stages when those marks have been removed.

We would rather not do further experiments involving inhibition of Bub1 and Haspin as this could perturb mitotic functions of these proteins beyond CPC recruitment and might complicate matters rather than clarifying them.

Post-review discussion:

"...we need some evidence that the levels, stability, and location of HP1 targeted to the chromatin by CENP-B tethering resemble what would be created by inhibition of H3S10 phosphorylation. Stability could potentially be accomplished by performing their FRAP assay from Figure 5 on EY-HP1alpha with Aurora B inhibition.

Expression levels and localization were already addressed as concerns in my review.

Second, to establish that the HP1 release is important, there needs to be some phenotype associated with it.

Currently, the only connection between the inability to release HP1 and a phenotype is the metaphase arrest.

Metaphase arrest is also the main assay for the first 4 figures. I therefore maintain that any issues with this assay are crucial to address." [ref 2]

We are slightly confused, as we believe that the metaphase arrest that we observe IS one strong phenotype and so is the continued activity of the CPC in H3S10 phosphorylation during G1 phase. We hope that our responses above show how we would work with the suggestions from the referee to address any issues with this assay.

"I agree that the demonstrating kinetochore-MT attachment instability is not necessarily novel/exciting, however this is the likely cause of the mitotic arrest. The authors seem to be concluding that constitutive tethering of HP1 leads to a defect in SAC signaling per se, rather than a defect in K-MT stability, which leads to an expected mitotic arrest through a properly-functioning SAC. I think it is important that they clarify this point." [ref 3]

We will address these concerns of the third referee by analysing the level of phosphorylated Dsn1 and cold stable microtubule assays as pointed out above.

2nd Editorial Decision

11 August 2017

Thank you again for sending your detailed point-by-point response, which I now had a chance to carefully go through. I find your explanations very helpful, and agree that they are able to clarify several of the major reservations that had arisen from the original reviews. I also appreciate your suggestion for an experiment in the spirit of referee 2's "additional comment", and your proposal to look at Dsn1 phosphorylation and microtubule assembly in response to referee 3's comments. Finally, I agree that additional focus on Haspin and Bub1 may not substantially improve but possibly rather complicate the interpretation of the study.

In summary, with these proposed changes and additional experiments included, we shall be happy to consider the manuscript further for re-review by the original referees, and I would therefore like to invite you to revise the manuscript accordingly and resubmit it using the link below. Should you have any additional questions in this regard, please do not hesitate to let me know.

1st Revision - authors' response

30 November 2017

We are pleased to re-submit our MS, EMBOJ-2017-97677, which has the new title "Heterochromatin protein 1 targets the CPC for activation at heterochromatin before mitotic entry" and an expanded list of authors: Jan G. Ruppert, Kumiko Samejima, Melpomeni Platani, Oscar Molina, Hiroshi Kimura, A. Arockia Jeyaprakash, Shinya Ohta and William C. Earnshaw. Jan and I have gone carefully over the many insightful comments of the referees. We were pleased that referees 1 and 3 were basically supportive of publication in the EMBO Journal, and hope that we are able to adequately address the various concerns of referee 2.

We have addressed all issues raised by the referees, and describe in our response below how we have done this. A few of the concerns were addressed by rewording or expanding explanations in the text, and we apologize if we were unclear in our original presentation of the work. As a metric of how seriously we have taken the comments of the referees, this revised MS now has 3 completely new main text figures, 3 new supplementary figures and 12 new panels in pre-existing figures

In order to improve the novelty of the MS, we have performed additional studies using chemical genetics, a panel of HP1 knockout cell lines, and live cell monitoring of histone modifications to probe the role of HP1 in the spatiotemporal regulation of H3S10 phosphorylation and the relative timing of H3S10ph and H3T3ph formation during mitotic entry. This new work was initially motivated by our observation that the CPC clustered at centromeres by tethered HP1 remains catalytically active throughout the cell cycle (as we have now shown conclusively) and was further inspired by the great suggestion of Referee 1 to do live-cell Fab tracking experiments. Indeed, we extended that experiment by double-label tracking of H3S10ph and H3T3ph in the same living cells. This led us to the interesting conclusion that HP1 targets the CPC to its sites of action well *before* H3T3 is phosphorylated. This leads to a significant addition to the paradigm for CPC targeting and activation in mitosis and the revelation that HP1 seems to be particularly good at creating a local environment that promotes Aurora B activation.

We attach below a detailed response to the comments of the referees. We will look forward to hearing from you when the secondary review process is complete.

Response to Referees:

Referee #1:

It has been long known that HP1 dissociates from the heterochromatin histone mark H3K9me3 upon Aurora B-dependent phosphorylation of H3S10 during mitosis, but its functional significance is largely speculative. By tethering HP1 to the centromere via CENP-B fusion, Earnshaw and his colleagues here now demonstrate that HP1 dissociation is required to allow the CPC to dynamically change its localization from the inner centromere to the spindle midzone during the course of mitosis. If HP1 is tethered to the centromere, INCENP, which is known to directly bind to HP1, is targeted to the centromere, leading to the stable localization of the CPC to the centromere throughout the mitotic progression and even in telophase and G1 phase cells, where H3S10 is remain phosphorylated at the centromere. Forced enrichment of HP1 by this method abortively activates the spindle assembly checkpoint and arrest cells in metaphase, consistent with a published observation on cells expressing INCENP tethered to CENP-B.

Overall, the authors performed the majority of experiments in a carefully controlled manner, and results support the authors' major conclusion. Although I have a modest reservation of authors' novelty claim, I support publication of this manuscript after minor revisions listed below.

We thank the referee for these thoughtful and positive comments and for supporting the publication of our manuscript in the EMBO Journal.

1. One of the major importance of this manuscript is to provide evidence that HP1 dissociation from mitotic chromosomes is required for dynamic localization change of the CPC. Although the authors indeed demonstrated this point by elegant experiments, I am not comfortable of the claim described in the last sentence of Discussion, "The suggestion that HP1 is an auxiliary member of the CPC during early mitosis for the first time reveals the importance of H3S10 phosphorylation and the methyl/phos switch on HP1 binding for ensuring the normal dynamics of CPC movements during mitosis."

Nozawa et al (2010) actually demonstrated that HP1 recruits INCENP to heterochromatin in interphase, and upon inhibition of Aurora B, INCENP is mislocalized to chromosome arms in a manner dependent on HP1-INCENP interaction (Fig. S2 and S5). These observations have already hinted the importance of HP1 dissociation in CPC localization control. I feel that it is appropriate to discuss these data, and soften the priority claim.

We thank the referee for pointing out the results from the supplementary data of Nozawa et al (PMID: 20562864) and we have now included these observations in our discussion, as they are supportive for our findings on HP1 regulation of CPC localisation. We have also softened our priority claim, as requested on p. 17.

"Overall, our findings are consistent with a recent suggestion that the HP1 α can function as an additional subunit of the CPC (Abe et al, 2016; Nozawa et al, 2010)."

Again, on p. 20 we state: "Indeed, ZM447439 treatment, which leads to diminished H3S10ph and H3T3ph in in mitosis, results in decreased localisation of HP1 and INCENP at centromeres, accompanied by an increased HP1-dependent chromosome arm localisation of these proteins (Nozawa et al, 2010)."

2. Figure 1. I assume that fixed cells were analyzed in this experiment, and cells were counted regardless of transfection levels. If so, it would be important to comment on transfection efficiency, although I understand that better live imaging analysis supported the authors' conclusion.

The referee noted correctly that in our initial experiments, in which the mitotic delay was identified, cells were analysed independent of the expression level of the transiently transfected proteins. To perform a more detailed analysis of the mitotic delay, we subsequently used live cell imaging in Fig. 2 and there, we reported the results separately for cells expressing low, medium or high levels of chimeric protein.

To clarify the experimental setup of the initial experiments, we have reported the transfection efficiency (~70%) in figure legend 1B:

"Frequency of mitotic HeLa cells 24 h after transfection with the indicated constructs (transfection efficiency ~70 %, judged by fluorescence microscopy)."

3. Fig. 6. I am curious to know if H3S10ph is maintained throughout the interphase beyond G1. Although the authors note that H3S10ph is normally observed in G2, robust phosphorylation can be seen just prior to prophase, and level of H3S10ph is almost undetectable during S and early G2 in cancer cell lines. See Hayashi-Takanaka et al 2009 (PMID: 19995936).

We thank the referee for pointing out this question about the robustness of the H3S10ph mark in interphase. We have collaborated with Hiroshi Kimura (communicating author of PMID: 19995936) and established his method of loading labelled Fab fragments into cells to follow histone modifications in live cells in our lab. Live cell imaging enabled us to clearly follow the H3S10ph mark across an entire cell cycle, conclusively demonstrating that it persists. In control cells, no mark was visible in interphase cells following mitotic exit and prior to its normal appearance in late G₂ cells.

We could also show that the persistence of the mark was dependent upon continued Aurora B activity, as inhibition of the kinase with a low level of ZM447439 (insufficient to abolish H3S10ph labelling in mitosis) caused the mark to rapidly disappear. Thus, it appears that the kinase tethered at centromeres retains its activity across an entire cell cycle. This is really interesting as phosphatase activity is supposed to dominate over Aurora B activity during interphase. We speculate that stable tethered HP1 foci may keep the CPC clustered in a way that allows Aurora B to out-compete the phosphatases.

On p. 18 we say: "Our data reveal that stably targeted HP1 α (CB-EY-HP1 α) can apparently recruit CPC clusters in which the kinase activity is maintained even in the presence of interphase levels of competing phosphatase activity."

These experiments are presented in **Videos 1 and 2** and summarised in the **new Expanded View Figures EV2 C-D. and EV3.**

Inspired by this success we went on to label a Fab fragment recognising H3T3ph, and used this for double tracking of H3S10ph and H3T3ph in living cells. This and immunofluorescence with these antibodies revealed clearly that H3S10ph is associated with HP1 foci long before H3T3ph appears (see also our response to Referee 3).

These results are presented in the **new Fig. 6** and in **Video 4**. See also the new section of Results:

"H3S10 phosphorylation precedes H3T3 phosphorylation in G2"

4. Page 19. "Thus, when the CPC is held at G1 centromeres by HP1, the Aurora B kinase activity is sufficient to counteract any conflicting phosphatase activity, possibly through continuous intermolecular self-activation at these clusters (Sessa et al, 2005)."

Also cite Kelly et al 2007 (PMID 17199039), which experimentally demonstrates the clustering-induced activation of Aurora B.

We thank the referee for highlighting the work of Kelly et al 2007 as an helpful addition to our discussion and we have included this in the manuscript. This comment also inspired the low-dose ZM447439 experiment which appears to clearly indicate that there is a close balance of kinase and phosphatase activity at the persistent centromeric Aurora B (and also for the natural H3S10ph foci that appear at the CDK-as arrest point in G₂).

We now say on pp. 17/18

"The CPC is well known to be activated by clustering (Fuller et al, 2008; Tseng et al, 2010; Wang et al, 2011; Kelly et al, 2007; Sessa et al, 2005)."

Referee #2:

In the manuscript "Heterochromatin protein 1 α dynamics regulate chromosomal passenger complex activity at centromeres", authors Ruppert, Earnshaw and colleagues engineer a variety of centromere-targeted HP1 α constructs and express them in HeLa and U2OS cells to determine the role of HP1 α release from chromatin in CPC regulation. The experiments demonstrate that constitutive targeting of HP1 α to centromeres results in a severe cell cycle delay that is dependent on the chromoshadow and hydrophobic pocket domains of HP1 α as well as the SAC. Although the authors create a few molecular tools with some potential, I feel that these tools were not used to sufficiently advance our knowledge of the role of HP1 α in CPC localization and function. Unfortunately, I do not believe that the conclusions drawn by the authors are of significant enough impact to be published in The EMBO Journal. In addition, I have many concerns about the nature of the experiments and the lack of controls as outlined below. I do, however, propose a

suggestion for how the authors could potentially use some of the tools that they have created to address a more pressing question that would likely be of high interest in the final section of this review.

We thank the referee for carefully going over our manuscript and hope that we have sufficiently addressed the various concerns with our extensive new experiments and re-writing.

Major concerns:

1. It is unclear where precisely CENP-B conjugation targets fusion proteins. The authors claim that it tethers HP1alpha to "its normal location in the inner centromere". However, CENP-B is generally considered to be a centromeric protein, not an inner-centromeric protein. Although the authors are correct in citing that CENP-B has been observed "throughout the centromeric region and beneath the kinetochore" in some cases, this does not necessarily qualify as "its normal localization".

Published experiments using immunogold labelling of CENP-B, which is generally accepted as state-of-the-art for localisation of an endogenous protein, show that the protein extends from the inner kinetochore down through the central domain of the centromere in HeLa cells (PMID: 2335558 – see also the double label fluorescence microscopy of endogenous CENP-B and CENP-C in PMID: 9382805). In our view, these careful controlled studies of the localisation of the endogenous protein are more reliable than numerous light microscopy studies of exogenously expressed CENP-B and CENP-B fragments. Our view is that CENP-B is located throughout the centromere. However, this is potentially complicated, because methylation of the CENP-B box can prevent CENP-B binding. Therefore, the precise localisation of CENP-B might differ between different cell lines or even different clones of the same cell line.

As discussed below, we agree with the referee that our initial statement that CENP-B DBD tethers HP1alpha to "its normal location in the inner centromere" was overly simplistic. In response to this concern, we have removed the statement about the "normal localization" of the tethered constructs (see below).

Evaluating potential differences between HP1alpha and CB-HP1alpha are made especially difficult by the lack of a direct comparison of localization for the CB-EY-HP1alpha and EY-HP1alpha constructs. This would preferably be on cells with lower amounts of over-expression. Additionally, line scans across the two kinetochores would be helpful to get a better idea of precise localization.

As suggested by the referee, we performed a detailed line scan analysis comparing HP1 tethered with wild type and mutant CENP-B-DBD to the distribution of untethered HP1 in **new panels in Expanded View Fig. EV1A,B**. We find that during prometaphase, when tension is low, the tethered HP1 constructs show an identical distribution to EYFP-HP1. Interestingly, at metaphase, when the centromeres are under tension, two peaks of the tethered constructs move outward along with the separating kinetochores (though a significant portion of the tethered HP1 remains in the central centromere). In contrast, the untethered HP1 remains as a single broadened peak in the central domain of the centromere. Interestingly, the distance from the kinetochore to the peak of the tethered HP1 remains very similar to what it was in kinetochores not under tension. Thus, the CENP-B-DBD tracks the behaviour of the kinetochore under tension, but does not approach it. Importantly, when we tether HP1 using a CENP-B DNA-binding domain, we see a phenotype extremely similar to what was seen by Lens and coworkers when they tethered INCENP using a similar CENP-B construct (PMID: 19150808).

In summary, our line-scan experiments fit with the notion that CENP-B is located throughout the centromere, with a significant population proximal – though internal – to CENP-C in the kinetochore.

In response to these new results, we have changed the text as follows on p. 7:
 "HP1 tethered using either the wild type or mutated CENP-B DBD colocalised with untethered EY-HP1 α in the inner centromere of prometaphase cells where centromeres are not under tension (Fig. EV1A). In metaphase cells, where centromeres are stretched, the tethered HP1 α split into two peaks that tracked the separating kinetochores, while untethered EY-HP1 α remained as a single, somewhat broader, peak in the inner centromere (Fig. EV1B). The tethered CB-EY-HP1 α remained 0.2 - 0.3 μ m internal to the peak of CENP-C, suggesting that it occupies a kinetochore-proximal domain, as previously described for CB-INCENP (Wang et al, 2011; Hengeveld et al, 2017; Liu et al, 2009)."

The complications with discerning between centromeric and inner-centromeric localization are highlighted by the recent publication Hengeveld et al. 2017 (DOI: 10.1038/ncomms15542). Initial

work published by the same group interpreted the chromosome missegregation phenotype of CB-INCENP-expressing cells as resulting from increased/constitutive activity near the kinetochore. However, their recent results suggest that the CB-INCENP construct actually moves the CPC from the inner-centromere to the centromere, resulting in a cohesion defect. To demonstrate that a similar unintended consequence is not occurring for the CB-EY-HP1 α construct, it would be good to demonstrate that the phenotype that they observe is not from cohesin disruption, perhaps by looking at chromosome spreads.

Hengeveld et al. 2017 and the preceding work by the Lens group (Liu et al. 2009) use the DNA binding domain of CENP-B fused to INCENP. In that very complex and elegant paper, when describing the localisation of their fusion proteins, the authors state “Although some Aurora B remained associated with centromeric heterochromatin in CB-INCENP expressing cells (our italics), we will refer to this CB-INCENP-shifted pool of Aurora B as ‘kinetochore proximal’”. Thus, the localization of Aurora B in their experiments is complex.

Nonetheless, we agree with their terminology that in metaphase cells, the CENP-B-DBD-HP1 could be referred to as “kinetochore proximal” (new Fig. EV1A, B).

Hengeveld et al. 2017 saw the cohesion defect referred to by the referee only when the endogenous INCENP was depleted and CB-INCENP was expressed (see their Figure 2B). However, in our experiments we do not deplete any endogenous proteins when assessing the effect of tethered HP1. Importantly, in our detailed analysis of live imaging of cells expressing CB-EY-HP1 we saw no evidence of a cohesion defect.

The above discussion reveals that this is a complex issue. We have made the following changes to address the referee’s concerns:

- **We have analysed the localisation of EY-HP1 compared to CB-EY-HP1 and included line scans as suggested by the referee (new Expanded View Figure EV1A,B)**
- **We re-phrased the HP1 tethering location on p. 7 as quoted in the answer to the previous point and using the terminology suggested by Susanne Lens (see text quoted in response to the previous point).**
- **We performed chromosome spreads, but do not include them as no cohesion phenotype was observed.**

2. To demonstrate the specificity of the metaphase delay phenotype for Aurora B over-activation, the authors look at the mitotic phases of CB-EY-HP1 α -expressing cells in the presence of the Aurora B inhibitor ZM447439 (Figure S4D). This is an absolutely essential control to demonstrate the specificity of the phenotype from perturbation of HP1 localization. Unfortunately, the shift in mitotic phase upon addition of the inhibitor appears to be dominated by the anaphase defects associated with Aurora B inhibition. 40-50% of cells appear to be in anaphase in the ZM treatment compared to less than 10% for controls (e.g. Fig. 1C). It is therefore impossible to determine if the decrease in metaphase cells is due to recovery of the spindle checkpoint arrest or due to the increase in anaphase and telophase cells. Minimally, measuring of the timing of metaphase following drug treatment would be necessary to distinguish between these two possibilities. Addition of a ZM-treated control without CB-HP1 α would also be helpful.

We apologise if the experiment was not described properly and caused confusion. In this experiment, we treated the cells with ZM for only 30 minutes before fixation. Therefore, the abundance of cells in anaphase and telophase must result from the sudden release of cells from the metaphase delay that occurred during the preceding 24 h and not from a gradual accumulation of cells blocked in mitotic exit. Many studies of the CPC show that when the complex is inhibited, the phenotype is not an accumulation of cells in ana/telophase, but instead an accumulation of binucleated cells due to cytokinesis failure.

To further address this issue, we performed an additional experiment using flow cytometry as had been done in the Lens lab (PMID: 19150808). When we took care to analyse only cells in mitosis (identified by MPM2 staining and not perturbed by a prior synchronisation in S phase) we saw the same trend reported in PMID 19150808, although the overall percentage of cells was lower. This may be because their experiment appears to have included a component of 4n cells that were actually G1 tetraploid cells following failed cytokinesis caused by ZM447439 treatment.

We performed two controls for this experiment: untransfected cells treated with ZM (as suggested by the referee) and cells expressing the CENP-B:HP1 mutant that cannot bind client proteins. Both controls gave comparable results (Fig. 3G).

Overall, the conclusion of our new experiment (see the new Fig. 3G) was that the inhibition of Aurora B results in a decrease in the mitotic index in cells expressing the tethered wild-type HP1.

On p. 10, we say “To further assess whether Aurora B activity causes the observed metaphase delay, we exposed CB-EY-HP1 α -expressing cells to the Aurora B inhibitor ZM447439 (Fig. 3G). Flow cytometry analysis revealed that this resulted in a decrease of the mitotic index to a level similar to that of control cells expressing CB-EY-HP1 α ^{W174A} and untransfected cells.”

3. Measuring the degree of expression for the fusion constructs relative to wild-type is essential to interpreting these results. Artificial mislocalization of a protein is already difficult to extrapolate back to its ordinary function. Artificial mislocalization combined with overexpression would make the results even more difficult to interpret. Although the authors do a good job of showing that the different mutant constructs are expressed at similar levels, it is unclear how this compares to the endogenous HP1. A western with an HP1 α antibody would help with this.

We were aware of this concern, and we initially sorted our live imaging results into three separate bins, based on whether cells were expressing low, medium or high levels of the chimeric proteins and reported the results separately (the criteria and method are reported in the Methods). When doing transient transfections, we believe that it is more accurate to judge the level of expression at the single cell level by fluorescence microscopy than to take the average for the culture as a whole.

In response to the suggestion of the Referee, we now include an immunoblot staining with an anti-HP1 α antibody in a new Fig. 1D. This shows that the EYFP-HP1 construct was expressed at levels similar to the endogenous protein, but that the CENP-B:EYFP:HP1 constructs were, if anything, expressed at a *lower* level than the endogenous protein.

On p. 6, we now state:

“All constructs were expressed at levels comparable to or less than that of endogenous HP1 (Fig. 1D).”

4. As with any artificially-targeting construct, it is imperative to establish if the observed phenotypes associated with the fusion are due to targeting the protein to the desired location, and not removing them from another location. This is especially important for proteins that are known to oligomerize, such as HP1, where the chimeric protein could strip the endogenous from an important location within the cell. The W174A mutant is a good control for this, as it eliminates the phenotype while presumably leaving the dimerization capabilities of HP1 intact. However, given the potentially high degree of overexpression of the fusion construct (see above), it still seems quite plausible that the endogenous HP1 would be affected through the PxVxL-binding motif. It would therefore be good if the authors could either demonstrate or cite clear differences between the CB-EY-HP1 α phenotype and an HP1 depletion phenotype.

Hopefully the referee no longer has this concern, since we have shown that the chimeric proteins are expressed at modest levels and are not highly overexpressed. We also thank the referee for appreciating the utility of using the W174A mutant as a control. We do have difficulties understanding how “endogenous HP1 would be affected through the PxVxL-binding motif”. Neither the CENP-B-DBD nor HP1 α contains a PxVxL motif and studies like that by Nozawa et al. show that HP1 dimerisation is not affected by the W174A mutation.

Furthermore, publications that deplete HP1 do not describe phenotypes like we show in our work. Examination of the literature reveals the following examples of phenotypes following HP1 depletion:

- 1. less tightly paired sister chromatids and delocalization of Aurora B**
- 2. overcondensed chromosomes in which the distance between the sister chromatids, both at arms and at centromeres, was clearly increased with respect to control chromosomes and the centromeric Aurora B signal was lost**
- 3. multipolar spindles**
- 4. pseudoanaphase with scattered chromosomes along the elongated spindle**
- 5. a bipolar spindle in which from one to five chromosomes had not yet congressed to the metaphase plate**
- 6. mitoses with increased merotelic attachments**

In chromosome spreads, we note, if anything, a tightening of the association between sister chromatids in cells expressing CB-HP1 (data not shown). The chromosome arms are not

over-condensed. We also did not notice difficulties with chromosome congression, as shown in the example of the live cell imaging experiment – where a well organised metaphase plate is formed within 12 minutes after NEB (Fig. 2A – 12 min). We therefore believe that the W174A mutant is a suitable control, as recognised by the referee.

Finally, and to directly address this point of the referee, we performed additional experiments during the MS revisions, using cells in which the HP1 α and HP1 γ genes were disrupted by CRISPR/Cas9 gene targeting (data is shown in the new Fig. 7A-C). In live-cell imaging, these cells passed through the cell cycle with normal timing (see Video 5, which shows a movie with HP1 α /HP1 γ DKO cells. Furthermore, the mitotic index of these cells was 4.8%, not significantly increased from the parental HeLa cell line. Thus, our tethering of CB-EY-HP1 does not phenocopy what is seen by a disruption of endogenous HP1 α and HP1 γ .

The new experiments are described in a completely new section of results:

“Loss of HP1 α and HP1 γ abolishes H3S10ph foci in G2 cells.”

5. The authors conclude in their abstract "Our results reveal that H3S10 phosphorylation is essential to release HP1alpha and allow CPC mobility during mitosis." It is unclear how this statement comes from the results in the manuscript, as there are no experiments that disrupt H3S10 phosphorylation and look at HP1alpha release.

We removed this statement from the abstract of our revised MS. In the text, we make it clear that this is a speculation based on our own work on the strong binding of HP1 to the CPC, the classic studies on the methyl/phos switch that displaces HP1 from chromatin (PMID: 16222244, PMID: 16222246) and on the supplementary data of Nozawa et al (PMID: 20562864) as suggested by referee no.1. The relevant statement in our abstract has been significantly rewritten in the light of our new results:

“Our results suggest that HP1 may concentrate and activate the CPC at centromeric heterochromatin in G2 before Aurora B-mediated phosphorylation of H3S10 releases HP1 from chromatin and allows pathways dependent on H3T3ph and Sgo1 to redirect the CPC to mitotic centromeres.”

6. One of the authors' other main conclusions is that "it may be useful to consider HP1alpha as an auxiliary member of the CPC during early mitosis". I fail to see how this is useful. Distinguishing between a complex "member" or "auxiliary subunit" vs. a strong interactor as previous demonstrated (e.g. Abe et al. 2016) seems largely semantic.

We thank the referee for pointing out the work from Abe et al who state in their abstract “we show that heterochromatin protein 1 (HP1) is an essential CPC component required for full Aurora B activity.” Indeed, our original suggestion was based in part on their studies and conclusions. In our revision, we have cited Abe et al for suggesting this idea first (p. 17): “Overall, our findings are consistent with a recent suggestion that the HP1 α can function as an additional subunit of the CPC (Abe et al, 2016; Nozawa et al, 2010).”

This point is useful because e.g. in structural studies where various laboratories have had difficulties in expressing the soluble CPC complex, it is possible that the addition of HP1alpha may allow the production of suitable samples for future study.

Minor:

The authors state that "the centromeric localisation of these mutants was indistinguishable from that of CB-EY-HP1 α containing wild type HP1 α (Fig. S2B-E)." This statement requires quantification of intensity and localization, as some of the images look quite different from each other (the I165E mutant appears to have decreased localization and more background).

We thank the referee for highlighting the discrepancy between Expanded View Fig. EV1E and the corresponding text. We apologise for our previous misstatement in the main text, which is now rewritten to remove the word “indistinguishable”.

The CB-EY-HP1-I165E mutant does localise to centromeres but shows much faster dynamics compared to the other CB-EY-HP1 constructs, presumably because it cannot dimerise. Our data suggest that this has a significant effect on the DNA binding by CENP-B (which is normally a dimer, whereas the CENP-B DNA binding domain on its own binds DNA as a monomer). Our experiments strongly suggest that CENP-B dimerization is required for stable binding of this protein at centromeres. For CB-EY, which consists of only the DNA-binding domain and EYFP without any fused protein, we measured a half time of recovery in FRAP experiments that was ~7 times faster than the binding dynamics of the chimeric CB-

EY-HP1 α construct and was very similar to the kinetics seen with CB-EY-HP1I165E, which does not dimerise. Thus, in our experiments, dimerization was provided by the HP1 moiety. This is why we believe that CB-EY-HP1-W174A is a better control, as that chimeric protein can dimerize, but cannot bind PxVxL client proteins. The FRAP analysis of CB-EY-HP1-I165E is not shown in the MS because we do not use the CB-EY-HP1-I165E construct for any further experiments.

We have rephrased the relevant text as follows (pp. 7,8):

“The localisation of tethered CB-EY-HP1 α W174A resembled that of CB-EY-HP1 α WT, but CB-EY-HP1 α I165E was slightly more diffuse (Fig. EV1E). FRAP analysis revealed that introducing the I165E mutation into CB-EY-HP1 α , which prevents dimer formation in HP1, results in a t_{1/2} of recovery of 8 s. This is similar to the t_{1/2} of CB-EY (6.8 s), which consists of only the DNA-binding domain and EYFP without any fused protein, and suggests that CENP-B dimerization is important for its stable binding to DNA.”

Similarly, the claimed differences in Aurora B localization in Figure S4A-C are unclear and require quantification.

We have removed these data which although highly reproducible, were difficult to quantitate. Instead we have now focused on the localisation of Aurora B during telophase as that is much clearer to see (see Fig. 4A)

Suggestion:

In the discussion, the authors write "Interestingly, no metaphase delay is seen when HP1 α is tethered by a CENP-BDBD mutant that has impaired DNA binding and causes a roughly 3-fold increase in HP1 α dynamics at centromeres." This could indeed be interesting in light of recent work that demonstrates a decrease in HP1 localization associated with CIN in cancer cell lines, and suggests that decreased HP1 at centromeric regions could be a cause of CIN. If the authors use the CBmut-EY-HP1 α tool to further test this hypothesis and determine if extra HP1 α activity rescues CIN, it would greatly increase the impact of the manuscript.

We thank the referee for this interesting suggestion. In fact, we began work on this experiment using the CIN line U2OS and had inconclusive initial results. Upon further consideration, two things led us to decide not to pursue this further. Firstly, the principal paper (from the Watanabe lab) that inspired this work has since been retracted. Secondly, in U2OS cells there is an extremely high background of mis-segregation events. This made the experiments very difficult to interpret and to assess the significance of what we were seeing. We therefore decided to increase the novelty of our MS by focusing on new approaches to look at the strong interactions between HP1 and the CPC and determine how HP1 influences CPC activation and behaviour in interphase prior to mitotic entry.

Referee #3:

The study here investigates a role for HP1 α in the regulation of Chromosomal Passenger Complex (CPC) dynamics at the centromere during mitosis. The authors constitutively tether HP1 α to centromeres by fusing it directly to the DNA binding domain of CENPB. This results in (1) a spindle assembly checkpoint (SAC)-mediated mitotic arrest, and (2) inhibition of CPC relocation from centromeres to the midbody at anaphase. This study nicely demonstrates that the status of HP1 α binding to chromatin is able to dictate the recruitment and eviction of the CPC from centromeres. My only concern is the authors' interpretation of the results regarding the role of the CPC in SAC activity. Upon constitutively tethering HP1 α to centromeres via CENPB, they observe a robust mitotic arrest that is Mad2-dependent. They go on to conclude that the mitotic arrest is due to perturbation of CPC dynamics which prevent the CPC from participating in its normal role in SAC signaling. A more likely explanation for the mitotic arrest is that when Aurora B kinase is constitutively localized to the centromere region, it continues to phosphorylate mitotic substrates which results in unstable kinetochore-MTs and an expected Mad2-dependent mitotic delay/arrest. The authors should check if this is the case by assessing kinetochore-MT attachment stability (cold-stable MT assay, level of k-k stretch, etc) and measuring the phosphorylation of known Aurora kinase B substrates.

We thank the referee for the supportive review and for highlighting a lack of clarity in our text. Indeed, we agree with the referee that the most likely explanation for the mitotic

delay is that constitutive Aurora B is destabilising Kt-MT interactions and eliciting a normal SAC response. We have now attempted to make this point more clearly.

Following the suggestion of the referee, we have now demonstrated that HP1 tethering does indeed result in a decrease in cold-stable microtubules (new Fig. 3C, D) and an increased phosphorylation of Dsn1 (new Fig. 3E, F).

The accompanying text (p. 10) reads:

“To understand the reason for SAC activation, we used a cold-stable microtubule assay to determine whether HP1 tethering results in impaired microtubule attachment to kinetochores. Indeed, tethering of wildtype HP1 resulted in a reduced density of microtubules attached to kinetochores as well as kinetochores lacking any apparent microtubule attachments (Fig. 3C). Measurement of the overall intensity of microtubules after cold treatment revealed a clear contrast between cells expressing CB-EY-HP1 α and control cells expressing CB-EY-HP1 α W174A or untransfected cells (Fig 3D). Thus, tethering of CB-EY-HP1 α decreases microtubule attachment to kinetochores.

We hypothesized that CB-EY-HP1 α tethering to kinetochores might recruit the CPC, which is well known to regulate kinetochore-microtubule interactions (DeLuca et al, 2006; Cheeseman et al, 2006). We indeed observed an increased level of phosphorylated Dsn1, an Aurora B substrate (Welburn et al, 2010), at kinetochores in metaphase cells expressing CB-EY-HP1 α compared to untransfected cells and cells expressing CB-EY-HP1 α W174A (Fig. 3E, 3F).”

Additional comment:

-The authors state that CENP-B-HP1 α tethering results in CPC recruitment to the chromatin independently of pH3 and pH2A. They should formally test this by inhibiting Bub1 and Haspin and assessing if CENPB-HP1 α is still able to recruit the CPC.

Inspired by this comment of the referee, we went on to perform a detailed analysis in synchronised and cycling cells of the relative timing and localisation of H3S10ph and H3T3ph in HeLa cells. The results were striking – H3S10ph clearly is present significantly before H3T3ph. We have shown this in both fixed images (in the new Fig. 6A,B) and in Video 4 (stills shown in new Fig. 6C).

H3T3 and H2AT120 are widely believed to play crucial roles in CPC recruitment to the centromere but have not been described to play a role in CPC localisation upon anaphase onset. However, with the tethered HP1 we observe a substantial amount of Aurora B remaining at centromeres even in telophase and later stages when those H3 and H2A phosphorylation marks have been removed. Thus, in view of the large number of other experiments done (including the live-cell analysis), we hope that we might be excused from this chemical inhibition experiment.

Our results contrast with a previous result from the Higgins lab (PMID: 15681610), and we believe that this is likely due to the different antibodies used. Hiroshi Kimura informed us that for efficient labelling of H3S10ph in interphase, it is essential to have an antibody whose binding is not affected by adjacent H3K9me3. We have used such an antibody here, and it may be that in the published work, the anti-H3S10ph was indeed affected by H3K9me3.

Post-review discussion:

"...we need some evidence that the levels, stability, and location of HP1 targeted to the chromatin by CENP-B tethering resemble what would be created by inhibition of H3S10 phosphorylation. Stability could potentially be accomplished by performing their FRAP assay from Figure 5 on EY-HP1 α with Aurora B inhibition. Expression levels and localization were already addressed as concerns in my review.

Second, to establish that the HP1 release is important, there needs to be some phenotype associated with it. Currently, the only connection between the inability to release HP1 and a phenotype is the metaphase arrest. Metaphase arrest is also the main assay for the first 4 figures. I therefore maintain that any issues with this assay are crucial to address." [ref 2]

Our goal was to create a synthetic physiological state with our tethering experiments. Thus, we aimed to stably localise HP1 at centromeres and ask how this affected CPC behaviour. We believe that we have shown conclusively that this tethering is capable of stably recruiting the CPC to centromeres, even in interphase and furthermore, that the CPC remains catalytically active.

Regarding the issue of a phenotype, we believe that the metaphase arrest that we observe is a strong phenotype and so is the continued localisation and activity of the CPC in

H3S10 phosphorylation during G₁ phase. Thus, HP1 tethering produces CPC phenotypes both in mitosis and in interphase. Our statement of the potential importance of HP1 release was only a hypothesis based on published work from other labs, which we believe to be interesting and possibly worthy of a follow-up study.

We have shown that targeting CB-EY-HP1 to centromeres in many respects phenocopies the targeting CENP-B:INCENP (PMID: 19150808). We observed a similar increase in mitotic index, dependency on the spindle assembly checkpoint, decreased microtubule stability as shown by the cold-stable assay (new Fig. 3C,D), and increased Dsn1 phosphorylation (new Fig. 3E,F). Because these observations are confirmatory of the earlier published data – though extended in our case, because we can conclude that HP1 must be recruiting the CPC – we have followed up in more detail our novel findings that tethering of HP1 to centromeres causes a stable recruitment of Aurora B that can maintain local phosphorylation of histone H3 throughout interphase.

In revision, we have significantly extended our study by employing chemical genetics and the use of HP1 gene disruption to look at the patterns of H3S10 phosphorylation in late G2 cells in the absence of the chimeric gene fusion (See new Figs 5-7 and Expanded View Figs EV3 – EV5). We believe that the resulting new data looking at H3S10ph and H3T3ph in G2 cells are novel and interesting, particularly as they suggest the need to expand the paradigm of how the CPC is recruited to centromeres.

"I agree that the demonstrating kinetochore-MT attachment instability is not necessarily novel/exciting, however this is the likely cause of the mitotic arrest. The authors seem to be concluding that constitutive tethering of HP1 leads to a defect in SAC signaling per se, rather than a defect in K-MT stability, which leads to an expected mitotic arrest through a properly-functioning SAC. I think it is important that they clarify this point." [ref 3]

As stated above, we apologise if our text was misleading. Indeed we do believe that it is *normal* SAC signaling that is responsible for the mitotic delay in cells expressing wild-type CB-EY-HP1 apparently because HP1 is strongly recruiting the CPC to centromeres. We describe above our experiments trying to clarify the role of the CPC in this delay by looking at the effect of HP1 tethering on cold stable microtubules and on Dsn1 phosphorylation (new Fig. 3C-F). We believe that the decreased microtubule stability, provides a reasonable mechanistic explanation for why the SAC would delay mitotic progression.

3rd Editorial Decision

21 December 2017

Thank you for submitting your revised manuscript for our consideration. It has now been seen once more by the original referees 1 and 2, whose comments are copied below. In light of these reports, we shall be happy to publish the paper in The EMBO Journal, pending a number of specific clarifications/discussions and addressing remaining technical issues, as raised by both referees. I would also invite you to once more answer through a point-by-point response letter to the remaining conceptual reservations of referee 2.

In addition, there are several important editorial points I would like to ask you to incorporate into the final version.

Once we will have received the modified files addressing the various editorial and referee points, we should hopefully be able to swiftly proceed with final acceptance and production of the manuscript. I look forward to receiving your final version.

REFeree REPORTS

Referee #1:

The revised version looks fine for publication at EMBO J.

One minor thing to be changed:

Include a reference below related to the observation that H3S10ph precedes H3T3ph.

<https://www.ncbi.nlm.nih.gov/pubmed/14987995>

In this manuscript, the authors described:

"Consistent with previous observations [17], double labeling with anti-M31 and anti-phospho H3 antibodies showed that serine 10 phosphorylation begins in the G2 phase around heterochromatic foci containing M31 (Fig. 4A, G2). At this point, a fluorescence signal with anti-phosphothreonine 3 antibodies could not be discerned (see also Fig. 2D)."

This is more or less an anecdotal statement, and Earnshaw's current manuscript is significant in establishing this, but I feel it is fair to cite this.

Referee #2:

In the revision to their manuscript, Ruppert, Earnshaw and colleagues have substantially updated their study with additional experiments and controls. The authors have thoroughly responded to the reviewers concerns and significantly improved upon the manuscript. The data are quite convincing and the experiments appear to have been conducted rigorously. However, I still have concerns about the novelty and potential impact of the paper as well as some minor technical concerns.

Novelty and impact:

There are now three key conclusions from the data presented in the paper. Firstly, targeting of HP1alpha near the kinetochore results in a metaphase delay. As the authors themselves note, this is very similar to what has been published for targeting INCENP, and therefore targeting a direct binding-partner of INCENP would likely result in similar phenotypes. The authors have therefore deemphasized this discovery in their updated manuscript to focus on the effects of forced-localization of HP1alpha to centromeres in interphase. Unfortunately, the mitotic result is also the only part of the study with a functional phenotype (metaphase delay). The other two parts only show differences in localization and speculate that these will impact function in some significant way.

The second key conclusion is that persistent localization of the CPC to centromeres in G1 results in continuous H3 phosphorylation. This suggests that phosphatase activity in G1 is not overwhelmingly high.

The third key conclusion, which has been added to the revised manuscript, is that HP1-dependent H3S10 phosphorylation by Aurora B begins in late G2, well before the CPC is targeted to chromatin by H3T3 phosphorylation. This is an intriguing result that strongly indicates a shift in how the CPC localizes to chromatin at different stages of mitosis. However, the authors construct a cell line deleted of HP1 alpha and gamma that almost completely eliminates the G2 loading of the CPC and they do not note and defects in chromosome compaction or segregation. The role and significance of targeting the CPC to chromatin in G2 is therefore completely unclear.

The authors do a good job of acknowledging these limitations in the manuscript and generally do not make any claims that are not supported by the data.

I feel that the things that I have stated above limit the impact of these results. However, there are certainly some well-performed experiments here that will help advance the field. This study should certainly be published somewhere. If the editor and the other reviewers feel that it is appropriate for The EMBO Journal, then I can certainly support that decision.

Technical concerns:

I find some of the statistical analysis performed in the manuscript quite confusing. For example, for different figures the authors use:

"Fisher's exact test followed by the Benjamini-Yekutieli multiple comparison test"

"Mann-Whitney test followed by the Benjamini- Yekutieli multiple comparison test"

"Fisher's exact test followed by the Benjamini-Hochberg multiple comparison test"

"Kolmogorov-Smirnov test"

Given the unusual diversity of the statistical tests used, a section in the methods clarifying which type of experiment requires each of these different statistical comparisons and why would be helpful.

It is also often unclear if the statistical analyses were performed per experiment or per measurement. For example, in figure 3F, it is unclear to me if the p values are based on the variability per kinetochore, per cell, or per experiment. Since the figure legends typically state that each experiment was performed three times, I would assume that the statistical significance for all of the figures was performed per experiment. However, it seems unlikely that such low p values in figure 3F would be obtained with an n of three and such a high standard deviation. Additionally, for this figure, why is the median shown instead of the mean?

For figure 3G, the figure legend states "One representative experiment from three repeats is shown." For numerical data, why not average the three experiments?

One of the main strengths of the paper is the quantification of the microscopy. This appears to be missing in some of the figures, most notably 7B and C, where only a single anecdotal example is provided for the described results.

Given that the results for the targeted HP1 are similar to INCENP, I think other researchers would also be interested to know if there are cohesion defects. I understand that the experiment is different since the endogenous protein was not depleted, but I still think that adding the chromosome spread results would improve the manuscript.

Similarly, the FRAP results for the I165E mutation are discussed in the paper but not shown. This will certainly lead to some confusion as the reader tries to find the figure for that data. Is there a reason this wasn't included?

2nd Revision - authors' response

24 January 2018

We have now addressed all of the additional comments of the referees as well as the comments that you yourself raised. In doing so, we have made a number of changes to the text and figures, and those are described in our response below.

Response to Referees:

Referee #1:

The revised version looks fine for publication at EMBO J.

We thank the referee for supporting the publication of our manuscript.

One minor thing to be changed:

Include a reference below related to the observation that H3S10ph precedes H3T3ph.

<https://www.ncbi.nlm.nih.gov/pubmed/14987995>

In this manuscript, the authors described:

"Consistent with previous observations [17], double labeling with anti-M31 and anti-phospho H3 antibodies showed that serine 10 phosphorylation begins in the G2 phase around heterochromatic foci containing M31 (Fig. 4A, G2). At this point, a fluorescence signal with anti-phosphothreonine 3 antibodies could not be discerned (see also Fig. 2D)."

This is more or less an anecdotal statement, and Earnshaw's current manuscript is significant in establishing this, but i feel it is fair to cite this.

We thank the referee for highlighting the work of Polioudaki et al. Their study used fixed cells and individual antibodies – not double staining. Nonetheless it certainly did provide preliminary suggestive evidence for what we have shown. We have included this reference in our manuscript in the following statement: “Our experiments using Fab fragments to monitor the

dynamic behaviour of histone modifications in living cells reveal that H3S10ph appears in G2 long before H3T3ph, as previously suggested using fixed cells (Polioudaki et al, 2004)." **(Indicated in blue text on p. 18)**

Referee #2:

In the revision to their manuscript, Ruppert, Earnshaw and colleagues have substantially updated their study with additional experiments and controls. The authors have thoroughly responded to the reviewers concerns and significantly improved upon the manuscript. The data are quite convincing and the experiments appear to have been conducted rigorously. However, I still have concerns about the novelty and potential impact of the paper as well as some minor technical concerns.

We thank the referee for carefully going over the revised manuscript and acknowledging the quality of our additional experiments and controls.

Novelty and impact:

There are now three key conclusions from the data presented in the paper. Firstly, targeting of HP1alpha near the kinetochore results in a metaphase delay. As the author themselves note, this is very similar to what has been published for targeting INCENP, and therefore targeting a direct binding-partner of INCENP would likely result in similar phenotypes. The authors have therefore deemphasized this discovery in their updated manuscript to focus on the effects of forced-localization of HP1alpha to centromeres in interphase. Unfortunately, the mitotic result is also the only part of the study with a functional phenotype (metaphase delay). The other two parts only show differences in localization and speculate that these will impact function in some significant way.

The second key conclusion is that persistent localization of the CPC to centromeres in G1 results in continuous H3 phosphorylation. This suggests that phosphatase activity in G1 is not overwhelmingly high.

The third key conclusion, which has been added to the revised manuscript, is that HP1-dependent H3S10 phosphorylation by Aurora B begins in late G2, well before the CPC is targeted to chromatin by H3T3 phosphorylation. This is an intriguing result that strongly indicates a shift in how the CPC localizes to chromatin at different stages of mitosis. However, the authors construct a cell line deleted of HP1 alpha and gamma that almost completely eliminates the G2 loading of the CPC and they do not note and defects in chromosome compaction or segregation. The role and significance of targeting the CPC to chromatin in G2 is therefore completely unclear.

The authors do a good job of acknowledging these limitations in the manuscript and generally do not make any claims that are not supported by the data.

I feel that the things that I have stated above limit the impact of these results. However, there are certainly some well-performed experiments here that will help advance the field. This study should certainly be published somewhere. If the editor and the other reviewers feel that it is appropriate for The EMBO Journal, then I can certainly support that decision.

We thank the referee for supporting the publication of our manuscript.

Technical concerns:

I find some of the statistical analysis performed in the manuscript quite confusing. For example, for different figures the authors use:

"Fisher's exact test followed by the Benjamini-Yekutieli multiple comparison test"

"Mann-Whitney test followed by the Benjamini- Yekutieli multiple comparison test"

"Fisher's exact test followed by the Benjamini-Hochberg multiple comparison test"

"Kolmogorov-Smirnov test"

Given the unusual diversity of the statistical tests used, a section in the methods clarifying which type of experiment requires each of these different statistical comparisons and why would be helpful.

We appreciate the referee's concerns about our use of different statistical tests. To simplify matters, we have redone much of the statistical analysis, limiting ourselves to two statistical tests. We use the Fisher's exact test for analysing data in the contingency table format and the Kolmogorov-Smirnov test for unpaired nonparametric data. We use the Benjamini-Hochberg procedure for all multiple testing corrections. This is considered the standard for controlling the false discovery rate. Because the previous and current statistical tests are both correct and quite similar, the changes did not result in different statistical results, except for comparing CB-EY-HP1 α and EY-HP1 α prophase cells in Figure 1C (change from not significant to one star significance). For clarification, we have added the following statement in a new brief Methods section "Statistical analysis: The Fisher's exact test was used for analysing data of the contingency table format. The Kolmogorov-Smirnov test was used for unpaired nonparametric data. The Benjamini-Hochberg procedure was used for all multiple testing corrections." (Blue text on p. 30.)

It is also often unclear if the statistical analyses were performed per experiment or per measurement. For example, in figure 3F, it is unclear to me if the p values are based on the variability per kinetochore, per cell, or per experiment. Since the figure legends typically state that each experiment was performed three times, I would assume that the statistical significance for all of the figures was performed per experiment. However, it seems unlikely that such low p values in figure 3F would be obtained with an n of three and such a high standard deviation. Additionally, for this figure, why is the median shown instead of the mean?

We were slightly confused by this comment, but now understand that confusion may have arisen because in these experiments statistics are being applied at two levels. First, we determined a mean value for each individual kinetochore (from its constituent pixels) and only then did we determine a median value for all the kinetochores.

For all our experiments we performed the statistical analyses per measurement, which we believe is the standard procedure for the kind of data we show. For the statistical analysis in Fig. 3F the mean Dsn1ph value for each individual kinetochore of metaphase cells (i.e. the mean of the values for all pixels corresponding to that individual kinetochore) was used. This is stated in the first sentence of the figure legend and we added the following sentence to improve clarity:

"Individual kinetochores were analysed and compared for 60 (CB-EY-HP1 α expressing and untransfected cells) or 58 (CB-EY-HP1 α W174A) cells, respectively." (Blue text on p. 41.)

Further we adjusted the axis label, now describing the Y axis as "Dsn1ph mean intensity per kinetochore A.U."

Some confusion may have arisen because when comparing kinetochores, we have shown the median and interquartile range rather than the standard deviation as we have stated in the second sentence of the figure legend. We use the median to represent those results because the data are skewed. Indeed, representing the data using the mean would have biased the representation in favour for our result, since the difference between CB-EY-HP1 α and the two controls would have appeared bigger, however this would not be the correct way to represent the results.

For figure 3G, the figure legend states "One representative experiment from three repeats is shown." For numerical data, why not average the three experiments?

We originally followed the style of data representation shown in an earlier Science publication. However, we agree with the referee that there is no good reason not to show the average of the three experiments and we therefore adjusted Figure 3G as suggested by the referee.

One of the main strengths of the paper is the quantification of the microscopy. This appears to be missing in some of the figures, most notably 7B and C, where only a single anecdotal example is provided for the described results.

We thank the referee for pointing this out as a strength of our work. Usually, where we have provided no specific quantification, only the indicated phenotype was detected. We did not include a statement about the quantification in Figure 7B, because in contrast to the H3S10ph signal shown in Figure 7A 3 and 4, only the diffuse localisation of Aurora B as shown was observed. To improve the clarity to the reader, we included the following statement in the

figure legend “All interphase HP1 α and HP1 γ double KO cells (2) exhibited a diffuse localisation of Aurora B.” (Blue text, p. 44.)

As suggested, we performed a quantification for Figure 7C and now include the result in the revised figure.

Given that the results for the targeted HP1 are similar to INCENP, I think other researchers would also be interested to know if there are cohesion defects. I understand that the experiment is different since the endogenous protein was not depleted, but I still think that adding the chromosome spread results would improve the manuscript.

We appreciate the referee’s interest in this result, but as we highlighted in our initial response to the referees and the referee now acknowledges, our experimental design was fundamentally different from the study that s/he is referring to. Unlike the study of Hengeveld et al. 2017, we did not deplete the endogenous protein. Our work is much more comparable to an earlier study by the Lens lab (Liu et al 2009), than to their recent publication (Hengeveld et al. 2017). In contrast to the recent study, which describes cohesion defects when the endogenous versions of the tethered proteins are depleted, in their earlier study no such defects were described. We cannot exclude that there may be minor effects on sister chromatid cohesion following tethering of HP1 α , but any effects are subtle and to raise this issue would detract from the focus of this MS on the regulation of CPC activation. We appreciate the interest of the referee, but do not feel that the addition of this interesting, but only tangentially related, observation would strengthen the overall flow of our paper.

Similarly, the FRAP results for the I165E mutation are discussed in the paper but not shown. This will certainly lead to some confusion as the reader tries to find the figure for that data. Is there a reason this wasn't included?

We omitted these constructs from the graph in Figure 1E for the reasons of clarity of the graph and to minimize overlapping curves. However, we agree with the referee that this could lead to some confusion, and we therefore added the additional data in a revised Figure 1E.

Corresponding Author Name: William C. Earnshaw

Manuscript Number: EMBOJ-2017-97677R1